



# Inverse modelling of Köhler theory - Part 1: A response surface analysis of CCN spectra with respect to surface-active organic species

Samuel Lowe[1, 2, 3], Daniel Partridge[3, 1, 2], David Topping[4, 5], and Philip Stier[3]

[1]Department of Environmental Science and Analytical Chemistry, Stockholm University, Stockholm, Sweden
[2]Bert Bolin Centre for Climate Research, Stockholm University, Stockholm, Sweden
[3]Atmospheric, Oceanic and Planetary Physics, Department of Physics, University of Oxford, Oxford, UK
[4]School of Earth Atmospheric and Environmental Science, University of Manchester, Manchester, UK
[5]National Centre for Atmospheric Science (NCAS), University of Manchester, Manchester, UK

*Correspondence to:* dan.partridge@aces.su.se

**Abstract.**

In this study a novel framework for inverse modelling of CCN spectra is developed using Köhler theory. The framework is established by carrying out an extensive parametric sensitivity analysis of CCN spectra using 2-dimensional response surfaces. The focus of the study is to assess the relative importance of aerosol physicochemical parameters while accounting for bulk-surface partitioning of surface active organic species. By introducing an Objective Function (OF) that provides a diagnostic metric for deviation of modelled CCN concentrations from observations, a novel method of analysing CCN sensitivity over a range of atmospherically relevant supersaturations, corresponding to broad range of cloud types and updraft velocities, is presented. Such a scalar metric facilitates the use of response surfaces as a tool for visualising CCN sensitivity over a range of supersaturations to two parameters simultaneously. Using response surfaces, the posedness of the problem as suitable for further study using inverse modelling methods in a future study is confirmed.

The organic fraction of atmospheric aerosols often includes surface-active organics. Partitioning of such species between the bulk and surface phases has implications for both the Kelvin and Raoult terms in Köhler theory. As such, the analysis conducted here is carried out for a standard Köhler model as well more sophisticated partitioning schemes seen in previous studies.

Using Köhler theory to model CCN concentrations requires knowledge of many physicochemical parameters some of which are difficult to measure in-situ at the scale of interest. Therefore, novel methodologies such as the one developed here are required to probe the entire parameter space of aerosol-cloud interaction problems of high dimensionality and provide global sensitivity analyses (GSA) to constrain parametric uncertainties. In this study, for all partitioning schemes and environments considered, the accumulation mode size distribution parameters, surface tension $\sigma$, organic:inorganic mass ratio $\alpha$, insoluble fraction $f_{insol}$ and solution ideality $\Phi$ were found to have significant sensitivity. In particular, the number concentration of the accumulation mode $N_2$ and surface tension $\sigma$ showed a high degree of sensitivity.

The complete treatment of bulk-surface partitioning is found to model CCN spectra similar to those calculated using classical Köhler theory with the surface tension of a pure water drop, as found in traditional sensitivity analysis studies. In addition,





the sensitivity of CCN spectra to perturbations in the partitioning parameters $K$ and $\Gamma$ was found to be negligible. As a result, this study supports previously held recommendations that complex surfactant effects might be neglected and continued use of classical Köhler theory in GCMs is recommended to avoid additional computational burden. In this study we do not include all possible composition dependent processes that might impact CCN activation potential. Nonetheless, this study demonstrates

the efficacy of the applied sensitivity analysis to identify important parameters in those processes and will be extended to facilitate a complete GSA using the Monte Carlo Markov Chain (MCMC) algorithm class.

As parameters such as $\sigma$ and $\Phi$ are difficult to measure at the scale of interest in the atmosphere they can introduce considerable parametric uncertainty to models and therefore they are particularly good candidates for a future parameter calibration study that facilitates a global sensitivity analysis (GSA) using automatic search algorithms.

**1  Introduction**

Atmospheric aerosols have an influence on the earth's radiation balance, and thus the climate and its evolution, through many feedback effects and processes. Aerosols can act to absorb and scatter solar radiation (the direct effect) (McCormick and Ludwig, 1967). In addition, aerosols are also capable of acting as Cloud Condensation Nuclei (CCN), defined as dry aerosols that provide viable nuclei for condensational growth of cloud droplets at a given supersaturation. CCN activity has the capacity

to have a influence on both cloud micro- and macro-physics and thus the radiative forcing of the atmosphere, the indirect effects (IPCC, 2013). The likelihood of a given aerosol being CCN active is a highly non-linear function of many parameters (McFiggans et al., 2006). At fixed liquid water path, an increase in aerosol concentration serves to increase CCN and Cloud Droplet Number Concentration (CDNC), thus reducing average droplet size and increasing albedo - the first (Twomey) indirect effect (Twomey, 1974). Consequently, the reduced average effective droplet radius restricts the formation of droplets large

enough to precipitate and is hypothesised to increase cloud lifetime, the second (Albrecht) indirect effect (Albrecht, 1989).

Cloud-aerosol interactions currently represent the largest uncertainty in current global radiative forcing estimates (IPCC, 2013). It is therefore necessary that we fully understand the physicochemical parametric dependence of CCN concentrations in order to help constrain such uncertainties. With an increased understanding of CCN activation, more accurate aerosol representations and CDNC parametrizations can be implemented in global climate models (GCM) (Fountoukis and Nenes, 2005;

Abdul-Razzak et al., 1998; Quinn et al., 2008). The high dimensionality and non-linearity of CCN activation as modelled by Köhler theory requires computational speeds currently unavailable to capture the full complexity of a true aerosol description and its evolution within a GCM. This computational burden is increased further when trying to represent the complex organic fraction of aerosols accurately. It is therefore of critical importance that the community ascertains which parameters and processes CCN activity is most sensitive to. In doing so, unimportant processes that add computational burden can be neglected

and computationally efficient droplet activation parameterizations and aerosol descriptions can be developed for GCMs.

CCN concentrations vary across different environments (Yum and Hudson, 2002) and therefore it should be expected that the importance of various physicochemical parameters may vary between different environments as both the aerosol size distribution and the parametric description of condensational growth are important in determining CCN activity (Lance et al.,





2004). It is clear that the importance of some physical and chemical properties are certainly expected to be greater than others (Quinn et al., 2008; Nenes et al., 2002). GCMs rely on grid resolved CCN concentrations and it is well understood that CCN activity is a strong function of the aerosol population's composition and size distribution (McFiggans et al., 2006).

CCN are defined to be aerosol particles larger than some critical size, referred to as the activation diameter, determined by
particle composition and ambient supersaturation. However, aerosol size distributions often possess steep gradients so a slight change in activation diameter can have a significant impact on CCN concentrations. For this reason, in the context of a climate study, it is insufficient to perform sensitivity studies solely on the point of activation; to arrive at robust conclusions, a study of CCN concentrations at all relevant supersaturations - a CCN spectrum - is preferred. Numerous studies have been conducted to examine the sensitivity of both the activation diameter and CCN concentrations with respect to the relevant physicochemical
parameter of the aerosol population (Fitzgerald, 1973; Roberts et al., 2002; Wex et al., 2008; Ervens et al., 2010). Such studies are instructive but are often restricted to perturbations in a single parameter, a one at a time analysis (OAT), also known as a Local Sensitivity Analysis (LSA), thus failing to probe the entirety of the full multi-dimensional parameter space. One way to improve on existing sensitivity studies is to embrace a novel inverse modelling methodology. The first application of inverse modelling to assess the effects of parametric uncertainty in aerosol-cloud interactions was undertaken by Partridge
et al. (2012), in which Markov Chain Monte Carlo (MCMC) simulation was used for inference of the posterior parameter distribution in a Bayesian framework. Here an inverse modelling methodology is developed for the analysis of CCN spectra to facilitate further study using MCMC simulation. The benefit of such a framework is twofold. Firstly, the inverse modelling framework facilitates a Global Sensitivity Analysis (GSA) that is able to probe the entire multi-dimensional parameter space, thus capturing any parameter interactions that can effect sensitivity estimates as seen by Quinn et al. (2008); Partridge et al.
(2011, 2012). Secondly, by introducing an Objective Function (OF), discussed below, a sensitivity analysis be carried out across all atmospherically relevant supersaturations simultaneously.

An inverse modelling framework not only enables the conditioning of parameter sensitivities on measurements, but also provides a method of diagnosing structural inaccuracies within models. Such inaccuracies present themselves as statistically significant discrepancies between optimised parameter values and their corresponding real-world observed values. In addition,
the technique also provides a method of parameter estimation for parameters which are not easily measured at the scale or in the context of interest. These advantages have led to the use of inverse modelling as a method of model calibration across a broad range of research subjects (Vrugt et al., 2004; Tomassini et al., 2007; Garg and Chaubey, 2010; Partridge et al., 2012; Wikle et al., 2013).

In this study, to the best of the author's knowledge, an inverse modelling framework is built for implementation in CCN
spectra modelling for the first time, where a CCN spectrum is defined as a number concentration distribution of CCN as a function of supersaturation. In order to diagnose the sensitivity of a function to parameter perturbations in a tangible way, an Objective Function (OF) is introduced. The OF provides a scalar metric by which the sensitivity of CCN spectra can be quantified with respect to both individual and multiple parameter perturbations simultaneously.

Before performing a GSA and full multi-dimensional parameter optimisation procedure using an automated search algo-
rithm, it is deemed judicious to first scrutinise the posedness of the problem in question (Pollacco and Angulo-Jaramilo, 2009;



Cressie et al., 2009). Should parameters be non-identifiable it may certainly be expected that multivariate statistical algorithms used to conduct parameter optimisation and GSAs may fail. To confirm that inverse of modelling CCN spectra is a well posed problem, response surfaces are invoked in this study as done by Toorman et al. (1992); Šimůnek et al. (1998); Vrugt et al. (2001); Partridge et al. (2011). Response surfaces are a graphical tool that enable investigation of the identifiability of pa-
rameters when considering CCN spectra susceptibility to aerosol perturbations in 2D planar subsets of the entire parameter space.

CCN activation of aerosols is modelled using Köhler theory (Köhler, 1936) by relating the equilibrium saturation vapour pressure ratio $s_{eq}$ at the particle surface to the wet radius $r_p$. An aerosol is deemed CCN active if the peak of the growth curve, the critical supersaturation $S_c$, is lower than the atmospheric saturation, thus allowing unstable growth in the presence
of sufficient water vapour. The original formulation was derived to describe the growth of a binary mixture of an inorganic salt with condensed water vapour but since then the theory has been subject to numerous developments to account for increasing levels of complexity to better represent aerosol systems observed in nature. These additions include, but are not limited to: multicomponent aerosols with concentration dependent organic acid solubility and surface tension (Shulman et al., 1996); addition of hygroscopic material via condensational depletion of soluble trace gases (Laaksonen et al., 1997); theoretical
derivation of an analytical solution for the point of activation when accounting for an insoluble core (Kokkola et al., 2008); inclusion of the bulk to surface partitioning of surface active organics (Sorjamaa et al., 2004; Topping, 2010); and the co-condensation of Semi-Volatile Oxygenated Organic Aerosol (SV-OOA) material (Topping et al., 2013).

One method of validating Köhler models is to carry out an aerosol-CCN closure analysis. Closure is achieved when model predicted CCN concentrations are within measurement uncertainties of data collected from CCN counters (CCNc) at a given
supersaturations. Numerous CCN closure studies have been performed with varying degrees of success (Bigg, 1986; Cantrell et al., 2001; Zhou et al., 2001; Broekhuizen et al., 2006; Bougiatioti et al., 2009; Martin et al., 2011). Broekhuizen et al. (2006) notes that aerosol-CCN closure is usually difficult to achieve and that such difficulty can be attributed to various sources of error including measurement biases in CCN chambers or particle size distribution measurements, or spatial and temporal variability during airborne measurements. In addition, they also note that studies unable to achieve closure were often those in which
organic carbon (OC) was more prevalent in the particle phase and were exposed to stronger influence from anthropogenic sources.

Recently, the importance of complex organic aerosols in CCN activation, and thus climatic forcing, has been observed (Lohmann et al., 2000; Jacobson et al., 2000; Chung and Seinfeld, 2002; Kanakidou et al., 2005). The organic fraction consists of thousands of different carbonaceous compounds with varying chemical and physical properties (Saxena and Hildemann,
1996). The organic fraction constitutes 20%-90% of atmospheric aerosol mass depending on the environment (Saxena and Hildemann, 1996; Jacobson et al., 2000; Putaud et al., 2004; Kanakidou et al., 2005; Zhang et al., 2007; Jimenez et al., 2009). In addition, Mircea et al. (2002) found the the presence of a water soluble organic carbon (WSOC) fraction could increase the number of CCN available in polluted regions by as much as 110%. Given this high sensitivity and the large range in organic mass fraction and associated chemical complexity, an importance must not only be placed the sensitivity
of modelled CCN concentrations to organic aerosol physicochemical parameters, but also on how that sensitivity interacts




with log-normal parameters describing the aerosol size distribution. This point calls for a more robust sensitivity analysis than typical individual parameter analyses, particularly when considering complex organics that require additional parameters that increase the problem dimensionality. Organic fraction constituents have changed dramatically from the pre-industrial period to present day, and thus the associated parameter ranges should be explored to investigate the influence this has had on our

climate.

There have been increasingly sophisticated formulations of Köhler theory, in relation to the organic fraction, developed over the last couple of decades in order to better model the role of organic aerosols in CCN activation. Sensitivity and modelling studies have reported that the presence of slightly soluble and surface active organic species can alter the point of activation for atmospheric aerosols (Shulman et al., 1996; Li et al., 1998; Sorjamaa et al., 2004; Henning et al., 2005; Topping, 2010;

Topping and McFiggans, 2012).

Facchini et al. (2000) suggested that the inability to achieve closure could be attributed to enhanced CCN activity due to accumulation of atmospheric polycarboxylic acids at the particle surface, thus depressing surface tension, as their molecular structure resembles that of HUmic LIke Substances (HULIS). In addition, Ekström et al. (2010) concluded that bio-surfactants have the capacity to possess a greater cloud-nucleating ability than even inorganic salts on account of measured surface tension

values below $30 \text{mNm}^{-2}$. HULIS, such as fulvic acids, depress surface tension (Li et al., 1998; Facchini et al., 1999, 2000), however more recently it has been acknowledged that the effect of bulk to surface partitioning must also be accounted for in the bulk aerosol properties (Sorjamaa et al., 2004; Topping, 2010) as depletion of solute from the bulk phase can produce a competing effect.

In reality, the transition between aerosol liquid and gas phases is not stepwise, i.e. the density profile is continuous rather

than a step function. To calculate the influence of bulk-surface partitioning organics, Sorjamaa et al. (2004) modified traditional Köhler theory to recalculate equilibrium curves in terms of bulk and surface quantities for binary and ternary mixtures. From their results, they deduced that surfactants may enhance growth of large droplets in the atmosphere thus decreasing cloud density. Topping (2010) derived an alternative theoretical basis that is able to model the effects of an unlimited number of surface active species and concluded that in order to have a comprehensive understanding of this phenomenon, model predictions

must be verified with CCN observations. Prisle et al. (2012) investigated the implications of bulk-surface partitioning for cloud droplet activation on a global scale using the ECHAM5.5-HAM2 global circulation model and recommend that an approach considering surface tension effects alone, and neglecting changes in bulk properties, is erroneous and should not be used. A full treatment of bulk-surface partitioning was found to return similar global CDNC results to a treatment neglecting bulk-surface partitioning altogether. Nevertheless, the topic still requires observational verification on all scales, and a GSA to probe the

entirety of the relevant multi-dimensional parameter space, in order to confidently arrive at correct conclusions.

In this study the effects of bulk-surface partitioning on CCN spectra are analysed using the model developed by Topping (2010) while simultaneously performing a response surface analysis of parametric sensitivity. To calibrate the sensitivity analysis, literature obtained best estimate parameter values are used to calculate a synthetic set of observations (calibration data). A response surface analysis of a suitably chosen objective function (OF) is carried out as performed by Partridge et al. (2011).

Parameters are chosen such that aerosol populations representative of marine average, polluted continental and rural continen-





tal air masses are considered, thus determining if certain environments require more sophisticated representations of organic aerosols than others.

## 1.1 Goals

The primary goal of this study is to increase our understanding of the parametric dependence of CCN spectra using Köhler theory to investigate the dependence of the cloud nucleating ability of aerosols to their physicochemical properties. By building the framework necessary for a complete GSA and parameter calibration, qualitative sensitivity information is provided in the form of response surfaces for simultaneous perturbations in two parameters. In addition to considerations of environmental dependent parameter sensitivities, the role of surface active organic compounds will also be explored to help constrain the parametric uncertainties introduced to climate models by the complex organic aerosol fraction. The specific questions to be investigated in this study are the following:

1. Is inverse modelling for parameter optimisation and a GSA of a CCN spectra model a well posed problem?

2. Qualitatively, how susceptible are CCN concentrations, across a range of atmospherically relevant supersaturations, to simultaneous perturbations in aerosol size distribution and physicochemical parameters

3. Does the bulk-surface partitioning of surface organics play an important role in CCN activity over an atmospherically relevant range of supersaturations, and how sensitive are the associated parameters?

4. How do the sensitivities in 2. and 3. vary across three distinct aerosol populations: Marine, rural continental and polluted continental?

## 2 Theoretical basis and materials

### 2.1 Multicomponent Köhler theory

The Köhler equation describes the equilibrium saturation vapour pressure ratio $s_{eq}$ of a condensible vapour at the surface of a wetted particle radius $r_p$,

$$s_{eq} = a_w exp\left(\frac{2M_w\sigma}{RT\rho_w r_p}\right) \tag{1}$$

where $M_w$ is the molecular weight of water, $\sigma$ is the surface tension of the wetted particle, $R$ is the universal gas constant, $T$ is temperature and $r_p$ is the particle radius (Köhler, 1936; Seinfeld and Pandis, 2012). The supersaturation $S_{eq}$ (%) is given by $S_{eq} = (s_{eq} - 1) \times 100\%$. The peak of the Köhler curve, in terms of supersaturation, the critical supersaturation $S_c$, defines the ambient supersaturation required for CCN activation.

The water activity term $a_w$ in Eq. 1 can be written in terms of an effective mole fraction $x_w^{eff}$,

$$a_w = x_w^{eff} = \frac{n_w}{n_w + n_s^{eff}} \tag{2}$$





where $n_w$ is the number of moles of water, and the effective number of moles of solute $n_s^{eff}$ can be calculated from the internally mixed Water Soluble Organic Carbon (WSOC) and inorganic components and their van't Hoff factors $i_i$ and $i_j$, respectively,

$$n_s^{eff} = \sum_{i=1}^{p} i_i n_i \chi_i + \sum_{j=1}^{q} i_j n_j \tag{3}$$

where the indices $i$ and $j$ span the number of organic ($p$) and inorganic ($q$) species. $n_i$ and $n_j$ are the numbers of moles of soluble organic and inorganic species respectively, and $\chi_i$ is the effective soluble fraction of the organic species (Shulman et al., 1996; Sorjamaa et al., 2004). In this study, organics will be assumed to completely dissolve, $\chi_i = 1$. Theoretically, the mathematical framework can treat a multi-component organic fraction, here however we choose to study just one, drop the summation and replace index $i$ with $org$ for clarity. Assuming a dilute solution, van't Hoff factors can be approximated by stoichiometric dissociation factors, $\nu_{org}$ and $\nu_j$, and the solution's osmotic coefficient $\Phi$ (Kreidenweis et al., 2005). $n_s$ can therefore be reformulated as

$$n_s^{eff} = \Phi \left[ \nu_{org} n_{org} + \sum_{j=1}^{q} \nu_j n_j \right] \tag{4}$$

For an aerosol of dry radius $r_d$ and insoluble fraction $f_{insol}$, the total number of moles of soluble substance can be re-expressed in terms of the organic fraction $f$ and individual inorganic component sub-fractions $\epsilon_j$ and each components molecular weight $M_{org}$ and $M_j$ and densities $\rho_{org}$ and $\rho_j$

$$n_s^{eff} = \frac{4}{3} \Phi \pi r_d^3 (1 - f_{insol}) \left[ f \frac{\nu_{org} \rho_{org}}{M_{org}} + (1 - f) \sum_{j=1}^{q} \frac{\epsilon_j \nu_j \rho_j}{M_j} \right] \tag{5}$$

As a final adjustment $f$ can be expressed in terms of the organic to inorganic ratio $\alpha$

$$n_s^{eff} = \frac{4}{3} \Phi \pi r_d^3 (1 - f_{insol}) \left[ \frac{\alpha}{1+\alpha} \frac{\nu_{org} \rho_{org}}{M_{org}} + \frac{1}{1+\alpha} \sum_{j=1}^{q} \frac{\epsilon_j \nu_j \rho_j}{M_j} \right] \tag{6}$$

The description of $n_s$ given in Eq. 6 is used for this study.

## 2.2 Bulk-surface partitioning

A brief overview of the theory behind the bulk-surface partitioning Köhler model developed by Topping (2010) and used here is given in this section. The reader is referred to Topping (2010) for a more detailed description. The mixing effects of multiple organic surfactants is not considered here.

The interface between bulk liquid and gas phases is not infinitely thin as Gibbs' surface thermodynamics would suggest (Sorjamaa et al., 2004); this surface phase of finite thickness is the region in which surface active organic compounds accumulate. In order to calculate the influence of the increased surface excess on the Kelvin (surface tension) and water activity terms simultaneously, a surface excess correction to the total molar quantity is required

$$n_{org}^s = n_{org}^t - n_{org}^b \tag{7}$$





where $n_{org}^s$, $n_{org}^t$ and $n_{org}^b$ are the surface excess, total and bulk molar quantities respectively. To separate the total quantity into bulk and surface quantities a solution of the Gibbs adsorption isotherm is required

$$n_{org}^s d\mu_{org} + Ad\sigma = 0 \tag{8}$$

where $A$ is the droplet surface area (m$^2$), $\mu_{org}$ the chemical potential of the organic and $\sigma$ is the surface tension (Nm$^{-2}$). The semi-empirical form of the Szyszkowski equation (Szyszkowksi, 1908) derived by Li and Lu (2001) is used

$$\sigma = \sigma_w - RT\Gamma^{wo} ln(1 + Ka_{org}) \tag{9}$$

where $\Gamma^{wo}$ (molm$^{-2}$), $K$ and $a_{org}$ are the saturated surface excess, adsorption equilibrium constant and activity of the organic compound, respectively, and $\sigma_w$ is the surface tension of a pure water. $\Gamma^{wo}$ is defined to be the molar excess of the surfactant in a unit surface area of the surface region over that in the bulk liquid region assuming the same number of moles of water in the two regions (Li and Lu, 2001). The superscript $wo$ refers to the position of the dividing interface being defined such that $n_w^s = 0$ or, equivalently, $\Gamma_w = 0$, and is dropped for notational convenience.

Solving Eq. 9 and 8, and assuming all activities can be represented by their effective bulk mole fractions, Eq. 2, the bulk mole fraction of the organic compound can be calculated from the roots of the quadratic equation,

$$(x_{org}^b)^2(A\Gamma K - n_w^t K - n_{org}^t K) + x_{org}^b(n_{org}^t K - n_{org}^t - n_w^t - A\Gamma K) + n_{org}^t = 0 \tag{10}$$

the solution of which is found using the standard quadratic formula and taking the negative root such that $0 < x_{org}^b < 1$ for a physical solution. Here the $eff$ superscript has also been dropped for notational convenience and all subsequent references to such variables will be to the effective values unless stated otherwise. Assuming the surface excess of water to be zero, the number of moles of surfactant in the bulk can be calculated as (Topping, 2010)

$$n_{org}^b = n_w^t \frac{x_{org}^b}{1 - x_{org}^b} \tag{11}$$

following this the water and surfactant activities can be calculated in terms of the bulk mole fraction of the surfactant

$$a_w = \frac{n_w}{n_w + n_{inorg} + n_w\left(\frac{x_{org}^b}{1 - x_{org}^b}\right)} \tag{12}$$

$$a_{org} = \frac{n_w\left(\frac{x_{org}^b}{1 - x_{org}^b}\right)}{n_w + n_{inorg} + n_w\left(\frac{x_{org}^b}{1 - x_{org}^b}\right)} \tag{13}$$

Equations 12 and 13 can be substituted into Eq. 1 and 9 to account for partitioning of the surfactant in Köhler growth curve.

With the partitioning described by Eq. 9, 12 and 13, there are four possible partitioning schemes for consideration. Application of the surface tension model, indicated by $\sigma^{nf}$, allows surface tension to be modelled as a function of organic activity and the empirically derived partitioning parameters, $K$ and $\Gamma$. This is in contrast to using a fixed value indicated by $\sigma^f$. Superscripts





$nf$ and $f$ indicate that the surface tension is not fixed and fixed, respectively. Accounting for the partitioning of the surfactant concentration to the surface phase is indicated by $a_w^p$ while assuming that the concentration remains solely in the bulk phase is indicated by $a_w^{np}$, where superscripts $p$ and $np$ indicate partitioning and no partitioning, respectively. The resulting four schemes are:

1. $a_w^{np}\sigma^f$

      2. $a_w^p\sigma^f$

      3. $a_w^{np}\sigma^{nf}$

      4. $a_w^p\sigma^{nf}$

Here 1. and 4. refer to simple Köhler theory and a complete treatment of bulk-surface partitioning, respectively. 2. and 3. refer
to schemes accounting for the partitioning of surfactant concentration to the surface phase and surface tension depression (eq.9) independently, respectively.

Modelling of CCN with Köhler theory involves many currently uncertain parameters, especially with respect to the organic aerosol fraction. In this study, the Köhler parameters probed in the sensitivity analysis will be $M_{org}$, $\rho_{org}$, $\Phi$, $\sigma$, $\Gamma$ and $K$ and compositional parameters $\alpha$ and $f_{insol}$. In addition, the log-normal parameters of the second (accumulation) mode, $N_2$, $\sigma_2$ and
$\bar{r}_2$, number concentration, geometric standard deviation and mean radius respectively, will also be probed as, in contrast to the first (Aitken) mode, a large from of accumulation mode particles are expected to be CCN active based on typical activation diameters. Thus the maximum dimensionality of the sensitivity analysis is 11, depending on the partitioning scheme used. In addition to the probed parameters, the following parameters are held fixed: $T = 285K$, $\nu_{org} = 1$, $\chi_{org} = 1$, $\nu_{\text{NaCl}} = 2$, $\nu_{(\text{NH}_4)_2\text{SO}_4} = 3$ and $\nu_{\text{NH}_4\text{NO}_3} = 2$.

To illustrate the impact of the different partitioning schemes on the CCN activation point, fig. 1 shows an example of critical supersaturation as a function of aerosol dry size for a mixture of NaCl and Suwannee River Fulvic Acid (SRFA) mass ratio of $\alpha = 2$. SRFA is prescribed values for its molecular mass, density and surface tension in solution of $610 \text{gmol}^{-1}$, $1570$ $\text{kgm}^{-3}$ and $55 \text{mNm}^{-2}$, respectively, based on measurements and results obtained by Dinar et al. (2006) and Taraniuk et al. (2007). These parameter values correspond to a single point of the multi-dimensional parameter explored. The activation point predicted by
classical Köhler theory $a_w^{np}\sigma^f$ (blue) using the surface tension of a pure water droplet is well replicated by the full partitioning scheme, $a_w^p\sigma^{nf}$ (magenta), for these particular parameters. The point of activation by classical Köhler theory $a_w^{np}\sigma^f$ using a fixed surface tension of $55 \text{mNm}^{-2}$ (black) that represents the influence of the surfactant on surface tension is well replicated by a partitioning scheme that accounts for a depleted bulk concentration of SRFA using the same fixed surface tension value $a_w^p\sigma^f$ (green). The partitioning scheme treating surface tension as concentration dependent Eq. 9, $a_w^{np}\sigma^{nf}$ (red) shows a more
complex relationship for the activation points. For smaller sized dry particles it more closely matches schemes using a depressed fixed surface tension value. At larger sizes however, it approaches the classical Köhler scheme using fixed surface tension of water; this regime change is attributed to a decreased surface:volume ratio for larger particles, thus reducing the influence of surface phenomena.





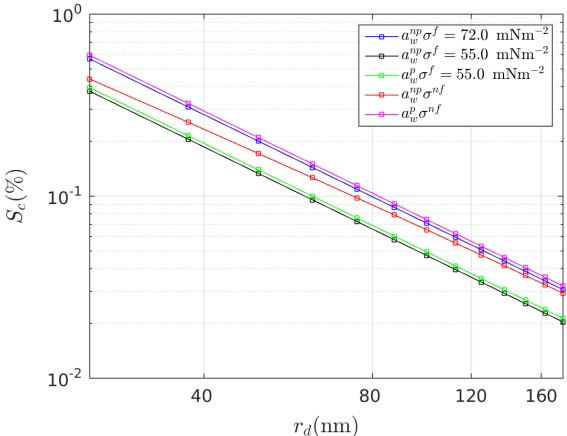

**Figure 1.** Critical supersaturation $S_c$ as a function of dry radius $r_d$ for all partitioning schemes. $a_w^p$ and $a_w^{np}$ labels indicate whether the partitioning effects are or are not accounted for in the water activity term $a_w$ respectively; $\sigma^f$ and $\sigma^{nf}$ indicate whether the surface tension is prescribed a fixed value or modelled using equation 9. Particles are internal mixtures of NaCl and SRFA with organic to inorganic ratio $\alpha = 2$. Complete solubility and ideality are assumed. The partitioning parameters for SRFA are $\Gamma$=0.0025 and $K$=35942.03.

### 2.3 Aerosol distributions and composition

In order to predict CCN spectra, Köhler theory must be coupled with an aerosol size distribution. Aerosol size distributions are well represented by a superposition of log-normal distributions (Seinfeld and Pandis, 2012)

$$n(r) = \sum_{i=1}^{m} \frac{N_i}{\sqrt{2\pi} \log \sigma_i} exp\left[ -\frac{(\log r - \log \bar{r}_i)^2}{2 \log^2 \sigma_i} \right] \tag{14}$$

5 where $N_i$, $\bar{r}_i$ and $\sigma_i$ are the log-normal parameters for the $i$th mode - number concentration, mean radius and standard deviation, respectively. In this study, only bi-modal distributions are studied, $m = 2$.

To analyse parameter sensitivity with respect to environmental aerosol characteristics, three distinct size distributions taken from existing literature:

   1. Marine average: Average global marine measurements from Heintzenberg et al. (2000).

10   2. Polluted continental: summertime air mass measurement from the Melpitz station, Germany (Birmili et al., 2001).

   3. Rural continental: SMEAR II station, Hyytiälä, Finland (Tunved et al., 2005).

Average distribution parameters, and their bounds, used for input in Eq. 14 are taken from the above references and are included in table 2. Distributions are generated over 400 logarithmically spaced dry radius bins ranging from 1nm to 5$\mu$m and shown in fig. 2. In this study the number concentration, mean radius and standard deviation of the second (accumulation) mode are considered for the sensitivity analysis and the first (Aitken) mode parameters are held fixed. The accumulation mode will be





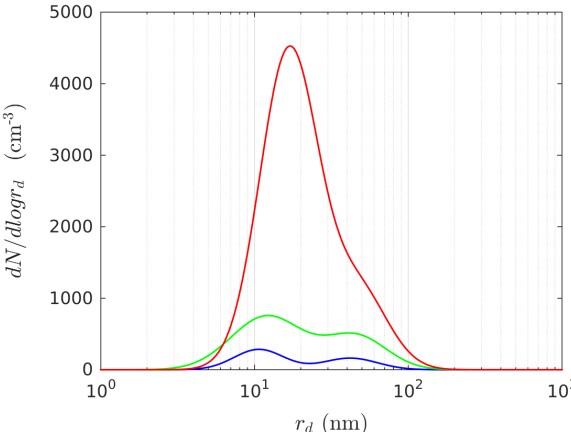

**Figure 2.** The marine average (blue), rural continental (green) and polluted continental (red) dry size distributions calculated using the true log-normal parameters given in table 2.

considered as typical activation diameters define a large fraction of the accumulation mode aerosol to be CCN active, thus a high degree of sensitivity is expected. In contrast, the smaller size of Aitken mode particles prevents them from being CCN active and therefore negligible sensitivity is expected.

In this study an aerosol composition representative of the ambient aerosol in each environment is prescribed. While the theoretical framework can treat external mixtures, for the sake of simplicity in this first study, the composition is specified as an internal, size-independent mixture of inorganic salts, a model organic (MO) surfactant and insoluble black carbon. Mass fractions of each component are taken from current literature estimates and are documented in table 1. Marine average composition is taken as an approximate average of measurements taken during high and low biological activity periods at the Mace Head atmospheric research station (O'Dowd et al., 2004); polluted continental from the Melpitz station, Germany (Poulain et al., 2011); and rural continental from non volatile mass measurements recorded in Hyytiälä, Finland (Häkkinen et al., 2012). The relative mass contributions are included in table 1.

The model organic used for this study has density $\rho_{org} = 1350$ kgm$^{-3}$ and molecular mass $M_{org} = 260$ gmol$^{-1}$ based on averages calculated from organic acids used in Topping (2010). The partitioning parameters for the MO were taken as the average of two strong surfactants, cis-Pinonic and Suwannee River fulvic acids, $K = 31071$ and $\Gamma = 0.00255$ mmolm$^{-2}$, to best capture the bulk-surface partitioning phenomena. In partitioning schemes that use a fixed depressed surface tension, take a value of $\sigma = 55$ mNm$^{-2}$ based on results found by Taraniuk et al. (2007) for humic-like substances. The insoluble black carbon component of the aerosol is modelled as elemental carbon with a density $\rho_{insol} = 2000$ kgm$^{-3}$ and molecular mass $M_{insol} = 12.0$ gmol$^{-1}$. The inorganic fraction is modelled as a mixture of salts, including ammonium sulphate $(NH_4)_2SO_4$, sodium chloride NaCl and ammonium nitrate $NH_4NO_3$ for each environment, the molecular masses and densities of which can be found in table 1.





## 2.4 Interpolation methods for CCN spectra modelling

To model the CCN number concentrations $N_{CCN}$ as a function of ambient supersaturation $S_a$, a grid ranging from 0% to 1.5% in increments of 0.01% is prescribed to capture the prevalent meteorological conditions that define various cloud types. Calculation of $N_{CCN}$ for a given $S_a$ requires computation of the activation radius $r_{act}$. For a given supersaturation, $r_{act}$ is

defined as the dry radius of the aerosol such that

$$S_c(r_d = r_{act}) = S_a \tag{15}$$

for a given internally mixed composition. Köhler curves are generated for each dry size class of the size distribution, as noted in section 2.3. In practice, owing to the discrete nature of the size classes, $S_a$ will be between two critical supersaturations $S_c^i$ and $S_c^{i+1}$ corresponding to a smaller and larger dry size $r_d^i$ and $r_d^{i+1}$, between which $r_{act}$ lies. A linear interpolation is employed

to calculate unique values of $r_{act}$ for each supersaturation. With $r_{act}$ determined, $N_{CCN}$ can be calculated by integrating the size distribution, however, as the size distribution is discrete it is calculated as a summation

$$N_{CCN} = \int_{r_{act}}^{\infty} n(r)dr = \sum_{j=m}^{400} N_j \tag{16}$$

where $q$ is the smallest size class activated and $n(r)$ is the number concentration size distribution function. It must be noted that $r_{act}$ will lie between the lower and upper bounds of the activated size class $q$. The practical difficulty this causes is twofold:

Firstly, when evaluating equation 16 as a summation, one must either discount the first bin number concentration or take its total number concentration. Secondly, should two or more $r_{act}$ values, corresponding to two or more $S_a$ values, lie in between the bounds of the same size bin, then non-unique calculations of $N_{CCN}$ will occur for different $S_a$, producing a step-like curve for the CCN spectrum. To circumvent this, fractional interpolation within the first activated size bin is employed between the upper and lower bounds, $r_q^u$ and $r_q^l$ respectively. Thus, in practice $N_{CCN}$ is calculated as follows

$$N_{CCN} = N^q \frac{(r_q^u - r_{act})}{(r_q^u - r_q^l)} + \sum_{i=q+1}^{400} N^i \tag{17}$$

The vector of $N_{CCN}$ values together with their corresponding $S_a$ values form the CCN spectrum.

## 3  Inverse Modelling Methodology

Inverse modelling is a methodology often used for parameter estimation and model calibration. The principal aim of inverse modelling is to find a set model input parameter values that produce model outputs that best represent measurement data.

The optimisation procedure is usually performed using a least squares or maximum likelihood criterion with respect to some objective function (see section 3.2) (Vrugt et al., 2006). Mathematically, it is formulated as follows. Let $\tilde{C} = \psi(X, \theta)$ denote the vector of $n$ model predictions, say CCN concentrations $\tilde{C} = (\tilde{c}_1, \ldots, \tilde{c}_n)$, where $\psi$ denotes the model and $X$ and $\theta$ are the input variables and parameters for optimisation, respectively. Given a vectorial set of observations $C = (c_1, \ldots, c_n)$, say



observed CCN concentrations, then the deviation of model predictions, for a given set of $\theta$, can be calculated as vector of residual concentrations $r(\theta)$

$$\boldsymbol{R} = \tilde{\boldsymbol{C}}(\theta) - \boldsymbol{C} = [(\tilde{c}_1 - c_1), \ldots, (\tilde{c}_n - c_n)] = [r_1(\theta), \ldots, r_n(\theta)] \tag{18}$$

Inverse modelling seeks a set of input parameters that minimises $\boldsymbol{R}$ but in practice minimising a vector quantity can be chal-

lenging. A solution to this problem is to introduce a scalar aggregate of the residuals called the Objective Function (OF). The aim now is to minimise this model-measurement discrepancy metric with respect to input parameter values. A parameter set that returns a zero valued OF corresponds to a perfect match between observations and model predictions. Producing a zero valued OF function with real-world observations is unlikely, however, synthetic modelling studies using model-generated synthetic measurements (c.f. section 3.2), such as this one, produce a zero-valued OF for parameter values used to generate the

synthetic measurements.

### 3.1 The Objective Function (OF)

In general, care should be taken when choosing the appropriate functional form of the OF. The functional form should reflect the characteristics of measurement errors seen in the relevant observation data set. A common definition for the OF is the simple least squares (SLS) or maximum likelihood estimator, valid when the measurement errors are believed to be both homoscedastic

and uncorrelated. The same assumptions are made here and a weighted Root Mean Square Error (RMSE) functional form for the OF is applied:

$$OF = \left[ \frac{1}{n} \sum_{i=1}^{n} w_i [\tilde{c}_i - c_i]^2 \right]^{1/2} = \left[ \frac{1}{n} \sum_{i=1}^{n} w_i r_i(\theta)^2 \right]^{1/2} \tag{19}$$

where $w_i$ is the weighting of the $i$th element. In this sensitivity study a framework is built for a future study that will use a complete GSA to provide a more rigorous analysis of important conclusions reached here. As such, as a first step, model-

generated synthetic measurements, henceforth referred to as calibration data (section 3.2), will be used in place of observations and weightings are set to unit, $w_i = 1$ for all $i$.

This simple objective function is chosen to establish the inverse methodology for the synthetic closure study. Future work will employ a more complex functional form of the OF that treats the heteroscedastic errors present in CCN spectra measurement data.

### 3.2 Synthetic calibration data

Real-world measurement data is normally used for parameter optimisation and model calibration. Here however, the inverse modelling framework is built up with transparency in mind and as such model generated synthetic measurements, calibration data, is used in place of observations. The calibration data is generated numerically using best estimate parameter values, henceforth referred to as the 'true' parameter values $\theta^{true}$ or calibration parameters. The calibration data is thus denoted



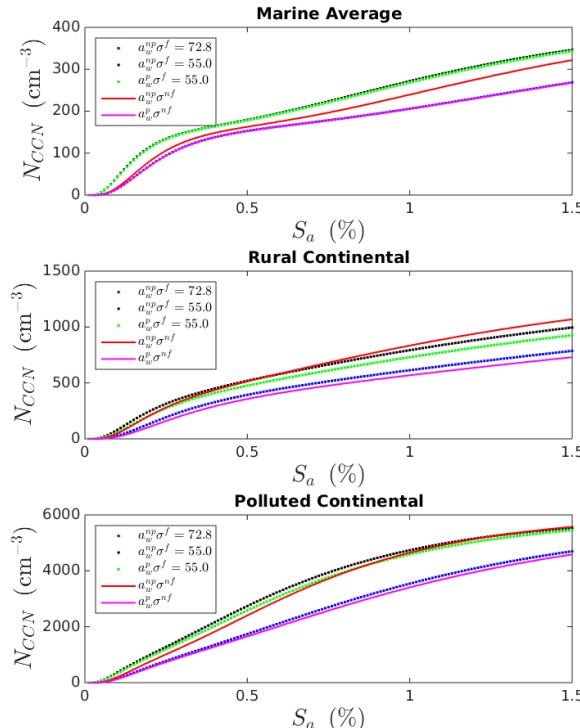

**Figure 3.** CCN spectra spectra calculated from true parameter values (table 2) - calibration data - for all environments and partitioning schemes. Partitioning schemes that use a fixed value for surface tension are shown by the circles, schemes using equation 9 are the lineplots. Individual parameter values are given in table 2.

$\tilde{C}(\theta^{true})$ and is a vector of CCN concentrations where each element corresponds to each point on a prescribed supersaturation grid ranging from 0.1% to 1.5%.

Appropriately generated calibration data is of critical importance to the inverse modelling methodology as the sensitivity of the CCN spectra is a function of the calibration parameters. The functional dependence of sensitivity on calibration parameters means that care must be taken when choosing calibration parameters so that reality is well represented and consistently between different environments. The parameters $\theta$ considered in this study are the Köhler parameters: $M_{org}$, $\rho_{org}$, $\Phi$, $\sigma$, $\Gamma$ and $K$; size distribution parameters of the second mode: $N_2$, $\sigma_2$ and $\bar{r}_2$; and compositional parameters $\alpha$ and $f_{insol}$. The true values of each of these parameters is discussed in section 2.3 and are documented collectively in table 2. The size distributions used in generating the calibration data for each of the three environments are calculated from the true log-normal distribution parameters and are shown in Fig. 2.

All 12 sets of calibration data generated from true parameter values for each partitioning scheme and environment are presented in Fig. 3, classical Köhler theory using the surface tension value of pure water (blue dotted curve) is also included





in the figure for reference. The differences between calibration data sets for different partitioning schemes arise for the same reasons as the changes in activation points shown in Fig. 1 and discussed in section 2.2.

## 4 Results and discussion

### 4.1 One at a time (OAT) parametric sensitivities

Typically, studies provide one at a time (OAT) sensitivity analyses of model outputs, e.g Wex et al. (2008). Although this methodology can be instructive it is not ideal. By performing an OAT analysis, large volumes of the full multi-dimensional parameter space remain unexplored and as a consequence the analysis misses important parameter interactions that could produce suppressed or increased sensitivity. A brief OAT analysis for a subset of the optimised parameters is included here as an instructive step in developing the response surface methodology. In addition, the OAT analysis facilitates the identification of supersaturations at which individual parametric sensitivities are greatest.

Consider a fractional perturbation $P$ to the true value of parameter $i$. The sensitivity $\delta C$ to the $i$th parameter is thus calculated as follows

$$\theta_i^{perturbed} = (1+P)\theta_i^{true} \quad P = 0.1 \tag{20}$$

$$\delta C = \tilde{C}(\theta^{true}) - \tilde{C}(\theta_i^{perturbed}, \theta_{j \neq i}^{true}) = (r_1, \ldots, r_n) \tag{21}$$

where $\theta_i^{perturbed}$ is the perturbed parameter for OAT analysis and the index $j$ runs over all other parameters. By applying the same fractional perturbation to each parameter the relative parametric sensitivities can assessed.

Figure 4 shows the calculated sensitivities for perturbations in $N_2$, $\alpha$, $\rho_{org}$, $\sigma$, $K$ and $\Gamma$ as a function of supersaturation. This is repeated for all partitioning schemes for the rural continental environment. It is clear that the surface tension $\sigma$ (green) is the most sensitive Köhler parameter when considering $a_w^{np}\sigma^f$ and $a_w^p\sigma^f$ bulk-surface partitioning schemes. This suggests that constraining uncertainties in $\sigma$ is important for CCN activation. At the scale of interest measurement of surface tension can prove challenging and inverse modelling could aid with this problem. For the simple Köhler case, $a_w^{np}\sigma^f$, the high degree of sensitivity to perturbations in $\sigma$ is in agreement with results obtained by Wex et al. (2008) which show a strong sensitivity of the critical supersaturation to $\sigma$ perturbations. CCN concentrations are also highly sensitive to the number concentration of the accumulation mode particles $N_2$ (black) at higher supersaturations: $|\delta C| \approx 40$ for convective supersaturations greater than 0.4% for all partitioning schemes. Large sensitivity to $N_2$ is expected as a large fraction of the accumulation mode particles have radii greater than that of typical activation radii, even at low supersaturations. In contrast, a reduced sensitivity would be expected of $N_1$ as the smaller Aitken particles are unlikely to activate at atmospherically relevant supersaturations. CCN concentrations are also sensitive to chemical and compositional parameters $\rho_{org}$ (blue) and $\alpha$ (red) but less so. There is symmetry in the sensitivity above and below the $\delta C = 0$ line for $\rho_{org}$ and $\alpha$, when considering simultaneous perturbations to





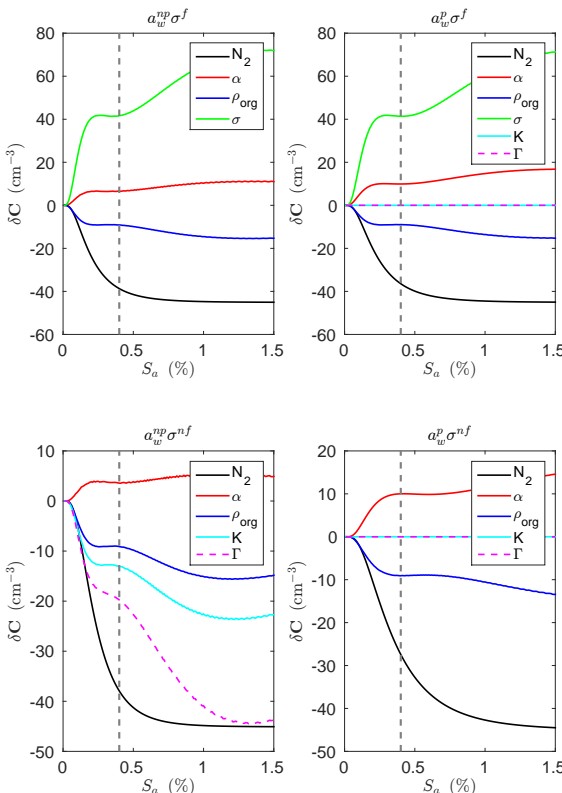

**Figure 4.** Sensitivity curves for rural continental CCN concentrations as a function of supersaturation. Selected parameters are perturbed individually by 10% for all partitioning schemes. In the upper and lower right panels curves for $K$ (cyan) and $\Gamma$ (magenta) almost overlap each other close to the $\delta\boldsymbol{C} = 0$ line. A vertical, grey dashed line is included at $S_a = 0.4\%$ to indicate a regime change between stratiform and convective cloud types.

these parameters it is likely there will be a non-unique set of parameter pairs returning minimal deviation from the calibration data due to parameter interactions.

For $a_w^{np}\sigma^{nf}$ and $a_w^{p}\sigma^{nf}$ partitioning schemes, $\sigma$ is replaced by the partitioning parameters $\Gamma$ and $K$ that are used to model the surface tension using Eq. 9. For $a_w^{np}\sigma^{nf}$ there is a strong sensitivity to both partitioning parameters suggesting that if these empirically derived parameters are to be used in Köhler modelling they must be known to a high degree of accuracy for meaningful conclusions to be drawn. For the full partitioning scheme $a_w^{p}\sigma^{nf}$ however, there is negligible sensitivity. The lack of sensitivity for the complete scheme is likely due to the competing effects of reduced surface tension and increased bulk water activity on the point of activation. It is necessary to analyse these two parameters simultaneously across their uncertainty ranges to provide a clearer picture, an instructive tool for such analysis is the response surface (c.f. section 4.2).





Global variability in updraft velocities has considerable importance for the aerosol indirect effect and leads to the development of different cloud types (West et al., 2014). In Fig.4 the distinction between stratiform and convective cloud types is illustrated by a grey, vertical dashed line at $S_a = 0.4\%$ corresponding to an updraft of approximately $0.5$ ms$^{-1}$ in marine environments (Chuang, 2006). It should also be noted that at this boundary between stratiform and convective clouds there is

transition in sensitivity to larger values for more convective systems. The supersaturation range used here is chosen to reflect the different updraft regimes associated with different clouds types. In Fig.4 we note that a local sensitivity maximum can be seen for most parameters, for all partitioning schemes, around 0.2% - 0.3% corresponding to stratiform cloud types. Furthermore, sensitivity is at a maximum for all parameters close to the upper limit of supersaturation 1.5%, these high humidities are often representative of deep convective systems typically seen in the tropics and give rise to Hadley cell circulation. There-

fore convective models such as the CRM-ORG developed by Murphy et al. (2015), with complex organic representations, require accurate input parameters for the description of organics and aerosols more generally for accurate simulation of CCN concentrations in deep convective systems.

## 4.2   Response Surface Analysis

Traditionally, sensitivity analyses have been performed on an individual parameter basis as in section 4.1. While these OAT

analyses can be instructive they are are inherently restricted by their inability to probe the full multi-dimensional parameter space. A GSA performs a comprehensive analysis that spans the entirety of the parameter space (Pérez et al., 2006). Such scrutiny of parametric sensitivity provides a more extensive and reliable set of results. This is particularly pertinent when applied to highly non-linear systems such as those found in cloud-aerosol interactions as multidimensional parameter interactions can significantly effect individual parameter sensitivities when the entire parameter space is explored e.g. Ervens et al. (2005);

Partridge et al. (2011, 2012).

In this section a graphical tool for qualitative CCN spectra sensitivity analysis for simultaneous perturbations in two parameters is employed - response surfaces. Traditionally, in 2D sensitivity analysis the surface represents a single parameter under investigation e.g. Quinn et al. (2008) who illustrated the changes in CCN concentrations at fixed superaturations to simultaneous perturbations in insoluble fraction and mean diameters. In what follows sensitivity is presented using response surfaces, as

employed by Partridge et al. (2011). Response surfaces are a graphical tool used to illustrate the behaviour of the OF - a scalar metric for deviation of model predicted CCN spectra from the appropriate synthetic observations - as a function of the perturbed aerosol physicochemical parameters. In this study, response surfaces are used to provide a qualitative analysis of CCN concentration parametric sensitivities across a high resolution range of supersaturations, thus sensitivity information for both stratiform and convective cloud types is captured by the OF. The response surfaces show how the OF varies in cross-sectional

planes of the parameter space and are thus instructive in testing the posedness of an inverse modelling framework as a method of CCN spectra parameter optimisation by providing some insight into how the OF may behave in the full parameter space. An inverse modelling framework can simultaneously facilitate a GSA and parameter optimisation when implemented using a Monte Carlo based automatic search algorithm methodology (Partridge et al., 2012). While the response surfaces only suggest how the OF may evolve when traversing the full parameter space, if the surfaces do not show a single well defined minimum





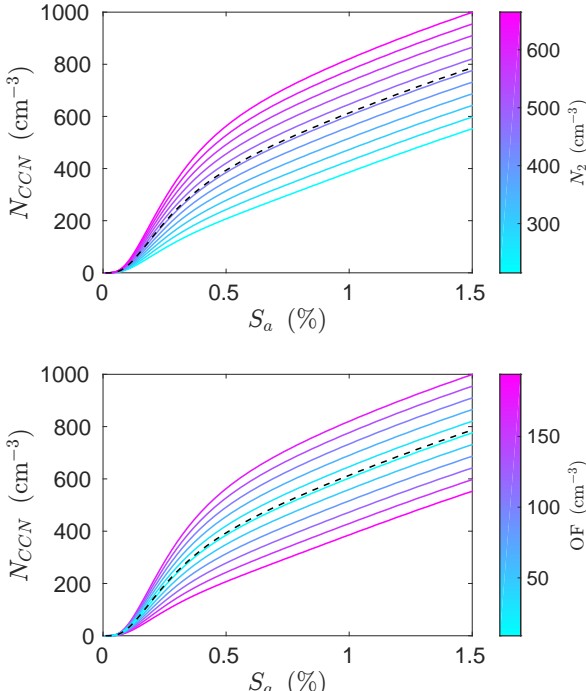

**Figure 5.** Rural continental CCN spectra for partitioning scheme $a_w^{np}\sigma^f$. In the top panel, the colour mapping indicates modelled CCN spectra as a function of $N_2$ within uncertainty ranges specified in table 2. In the bottom panel, the colour mapping indicates variation in the OF between the modelled spectra and the calibration data (dashed black line) for the corresponding calculations with respect to $N_2$.

then it may certainly be expected that inverse parameter optimisation may be unsuccessful (Partridge et al., 2011). Parameters that can take a large range of values while maintaining minimal deviation in CCN spectra from the calibration data are deemed 'non-identifiable' and will be difficult to calibrate based on the current information content of the OF. Response surfaces provide a way of visually discerning such parameters to be removed from such optimisation methods. Surfaces possessing a well

5  defined minimum are preferred as parametric search algorithms iterate more efficiently if the gradient of improvement points toward a single attractor within search space. Response surfaces containing single attractors and steep gradients suggest that the associated parameters are both sensitive and viable candidates for calibration. In addition, this high degree of sensitivity also implies that it is important to represent such parameters well in GCMs.

Using response surfaces to visualise the evolution of an OF across 2D parameter planes has been used effectively in similar

10  highly non-linear atmospheric inverse problems (Partridge et al., 2011). The upper panel of Fig. 6 shows a response surface of the critical supersaturation $S_c$, for a dry aerosol with a 75 nm radius and marine average composition, as modelled using traditional Köhler theory. The response surface provides a visual representation of the sensitivity of $S_c$ to simultaneous per-





turbations of $\alpha$ and $\sigma$. It is clear from this figure that sensitivity to $\sigma$ is greater than that of $\alpha$. In addition, non-unique values of $\alpha$ can result in the same $S_c$ value, this result is similar to results obtained by Wex et al. (2007) for a bulk parameter of chemical properties. While these results are useful, they do not test the entire distribution of aerosols at all atmospherically relevant supersaturations. Here we improve on such sensitivity analyses by analysing an objective function (OF) with respect

to CCN spectra. In the lower panel of fig. 6 is the response surface of the OF for the CCN spectra obtained by coupling Köhler theory with the entire aerosol population, for the same parameter perturbations. Blue crosses indicate the coordinates of the true parameter values in the planar subset of the the full parameter space. Sensitivity of the activation point has been performed numerous times in existing literature, it is evident, however, from fig. 4, that the parametric sensitivity of CCN concentration can vary depending on the atmospheric supersaturation. Therefore, by replacing $S_c$ with the OF discussed in section 3.1 as a

sensitivity metric across a range of supersaturations and coupling Köhler theory with the aerosol size distribution, a response surface analysis of CCN concentrations at all atmospherically relevant supersaturations is carried out simultaneously.

To further illustrate the methodology Fig. 5 shows how the rural continental CCN spectrum varies as $N_2$ is perturbed. In the upper panel the curves are coloured according to the perturbation in $N_2$, and in the lower panel they are coloured according to the value of the OF, or according to the magnitude of deviation from the calibration data generated from the true parameter

values (black dashed line).

All parameters of interest will be perturbed across a range of values that reflect uncertainties found in existing observations that include both laboratory and in-situ measurements. These ranges are documented in table 2. Parameter ranges for distribution parameters, $\alpha$ and $f_{insol}$ are deduced from references contained in section 2.3. The density and molecular mass of the surfactant are perturbed between minimum and maximum values of the five compounds studied in Topping (2010), while the

partitioning parameters $K$ and $\Gamma$ are perturbed between the values of two strong surfactants, cis-Pinonic and Suwannee River fulvic acid, Topping (2010). Surface tension was allowed to vary between 30 mNm$^{-2}$ and 72.8 mNm$^{-2}$ (pure water) to account for particularly strong surfactants such as bio-surfactants (Ekström et al., 2010). The effect of unideal solutions is also explored by exploring spectra sensitivity to perturbations in $\Phi$ between 0.75 and 1.0.

In what follows, parameter sensitivities for all four partitioning schemes in the marine average environment are analysed in

sections 4.2.1 - 4.2.4 before considering some environmental dependencies in section 4.2.5. Focus is given to the marine cloud CCN due as their extensive spatial coverage, high contrasting albedo relative to the surface (Warren et al., 1986, 1988) and long synoptic lifetime (Brenguier and Wood, 2009) result in a greater climate sensitivity than perturbations in other environmental CCN concentrations.

Blue crosses indicate the true parameter values, and collectively correspond to the full true parameter set $\theta^{true}$, and therefore

return a zero value OF. Constant value OF contours and a colour mapping are used to visualise the deviation of CCN spectra from the calibration data. Dark (hot) regions of the parameter plane indicate a high value OF (eq.19) and thus large deviations of modelled CCN spectra from the calibration data shown in Fig. 3. Light (cold) regions indicate low value OF returns and thus small deviations of modelled CCN spectra from the calibration data.



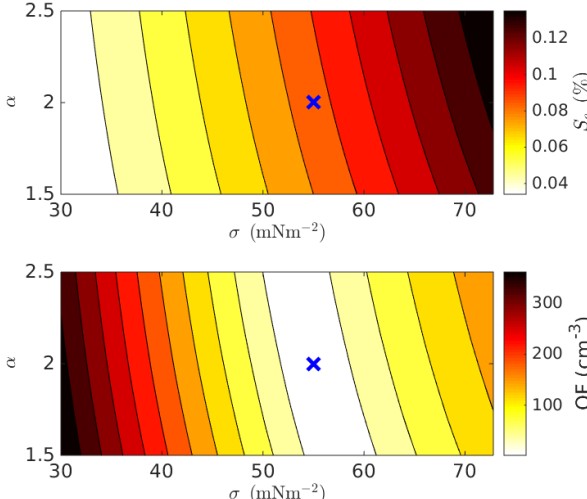

**Figure 6.** Top panel: Response surface for the critical supersaturation $S_c$ of a $r_d = 75$nm marine average aerosol for perturbations in $\alpha$ and $\sigma$ across ranges given in table 2. The blue cross indicates the true values of the two parameters. Bottom panel: Response surface for the OF as applied to marine average CCN spectra for the same parameter perturbations.

### 4.2.1  Classical Köhler theory: $a_w^{np}\sigma^f$

Figure 7A-D shows response surfaces for four parameter combinations for classical Köhler theory $a_w^{np}\sigma_{ws}^f$ in the marine average environment. This particular formulation of Köhler theory, when coupled with the size distribution parameters of the second mode, presents 9 parameters for analysis, leading to the calculation of 36 response surfaces, 13 of which possess a well
defined minimum. The response surfaces presented in fig. 7 are chosen to illustrate the relative sensitivities of size distribution and Köhler theory parameters with higher sensitivities. The complete set of response surfaces, not included, indicates that $\sigma$, $\alpha$, $f_{insol}$, $N_2$ and $R_2$ are the most sensitive parameters. In fig. 7D the response surface for perturbations in $\alpha$ and $\sigma$ does not contain a well defined minimum as seen in fig.7A-C. Interactions with $\sigma$ allow $\alpha$ to take any value across its uncertainty range and return a zero OF for a narrow band of $\sigma$ values close to its true value. This minimal deviation in CCN concentrations from
the calibration data identifies $\alpha$ as a non-identifiable parameter on this response surface.

### 4.2.2  Redistribution of surfactant concentration: $a_w^p\sigma^f$

Response surfaces were recalculated for the inclusion of bulk-surface partitioning effects in the Raoult term $a_w^p\sigma_{ws}^f$ which accounts for the reduced bulk concentration of surfactant when calculating the water activity. This allows the effect of bulk-surface partitioning on the bulk water activity to be isolated from the effects of a concentration dependent model of surface
tension, equation 9. In order to model the effects on the bulk activity the partitioning parameters, $\Gamma$ and $K$, must be introduced. Thus all 11 parameters are analysed in this setup, totalling 55 response surfaces 13 of which contain a well defined minimum.





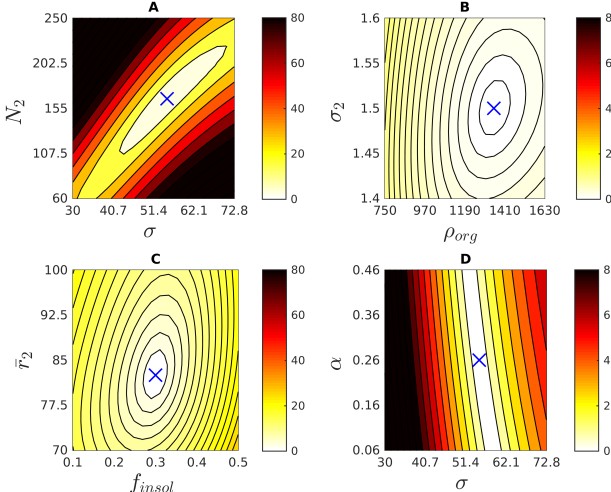

**Figure 7.** Response surfaces for $a_w^{np} \sigma^f$ in the marine average environment. Blue crosses indicate the true parameter values $\theta^{true}$ used to calculate the calibration data. The colour scale represents the value of the OF calculated for the modelled spectra against the calibration data for parameter values across the uncertainty ranges (table 2). Contours identify constant OF values on the surface (note different scales).

Figure 8 shows response surfaces that include the partitioning parameters. Response surfaces for parameter pairs common to this scheme and classical Köhler theory showed negligible changes suggesting this effect is relatively small when determining CCN spectra. This result is further highlighted in Fig. 8D, while parameter interactions similar to those seen in Fig. 7D are present, the scale of the sensitivity has been reduced by 7 orders of magnitude, meaning that, within the specified uncertainty

range, values of the partitioning parameters do not require to be known any more accurately for this partitioning scheme. The lack of sensitivity to the partitioning scheme as well as the partitioning parameters is attributed to its action solely through the water activity term. The water activity, in its mole fraction form, Eq. 2, is typically close to unity at the point of activation as $n_w \gg n_s$ and therefore any changes to bulk concentrations of the solute moles $n_s$ may certainly be expected to have negligible influence on the mole fraction when all other model features and parameters are held fixed. This is further reinforced by the

black ($a_w^{np} \sigma_{ws}^f$) and green ($a_w^p \sigma_{ws}^f$) curves in figures 1 and 3, wherein only small changes to the critical supersaturation $S_c$ and the CCN spectrum respectively are seen. In addition, this scheme should not be considered as an accurate representation of what occurs in nature; here a bulk concentration dependent water activity has been considered whilst applying a fixed surface tension approach that does not depend on a surface concentration of partitioning surfactant. Nevertheless, it remains instructive to isolate and ascertain the magnitude of the effect of such a phenomenon on CCN activation so that it can be disregarded in

future studies and model and parametrization developments.

The sensitivity to perturbations in solution ideality is also explored in fig. 8B and is found to have a similar sensitivity to the modal radius $\bar{r}_2$. Information on ideality of atmospheric particles is challenging to measure in-situ and given the relatively high sensitivity of the parameter shown here, it is a particularly good candidate for optimisation towards a unique value that corresponds to a best fit CCN spectrum using inverse methods. The chemical properties of the surfactant, $M_{org}$ and $\rho_{org}$



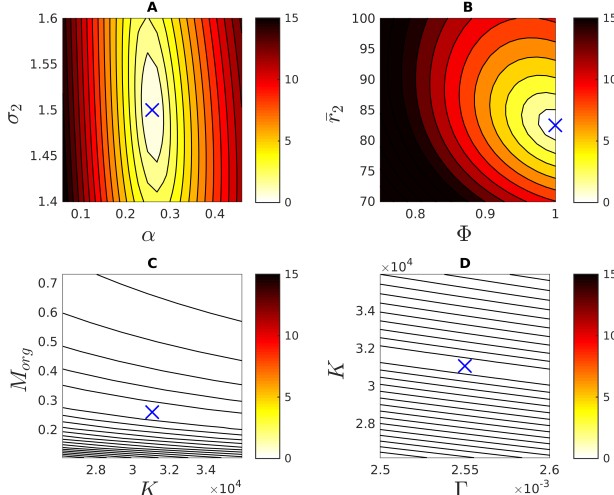

**Figure 8.** Response surfaces for $a_w^p \sigma^f$ in the marine average environment. Blue crosses indicate the true parameter values $\theta^{true}$ used to calculate the calibration data. The colour scale represents the value of the OF calculated for the modelled spectra against the calibration data for parameter values across the uncertainty ranges (table 2). Contours identify constant OF values on the surface (note different scales).

(figures not included), were found to be relatively insensitive when compared with other parameters for this partitioning scheme as well as classical theory in section 4.2.1. The relative unimportance of these chemical properties is in-line with the general conclusion reached by Dusek et al. (2006) that chemistry is less important than size. However, in contrast, the sensitivities of compositional parameters, $\alpha$ and $f_{insol}$, and the ideality of the solution $\Phi$ exhibit similar sensitivities in determining the CCN

concentrations as those of the size distribution parameters $\sigma_2$ and $R_2$. Dusek et al. (2006) carried out their sensitivity analysis on CCN size distributions at individual supersaturations of $0.25\%$, $0.4\%$ and $1.0\%$ thereby missing sensitivity information for all stratiform cloud types below $0.25\%$ and convective cloud types above $1.0\%$. Here the analysis has been carried out over a highly resolved range of atmospheric supersaturations. This difference in methodology is likely the cause of contrasting results as the discrete nature of their analysis may miss peaks in individual parameter sensitivities such as those seen in Fig. 4, the

effect of which are quantified and accounted for in the OF methodology developed here.

### 4.2.3 Surface tension considerations: $a_w^{np} \sigma^{nf}$

Here the effects of a concentration dependent surface tension, Eq. 9, are accounted for while the modifications to the water activity discussed in the previous section are neglected - $a_w^{np} \sigma^{nf}$. In this partitioning scheme the partitioning parameters, $\Gamma$ and $K$, replace surface tension so the analysis covers 10 parameters and thus 45 parameters planes, 21 of which possess a well

defined minimum.

Again, the surfaces show the same chemical parameters to be least sensitive. However, the CCN spectra show a higher degree of sensitivity to changes in $\Gamma$ and $K$ than in the $a_w^p \sigma^f$ case, confirming results from the OAT analysis. This result is expected on account of their action through the surface tension - the Köhler parameter commonly found to be one of the most sensitive, e.g.




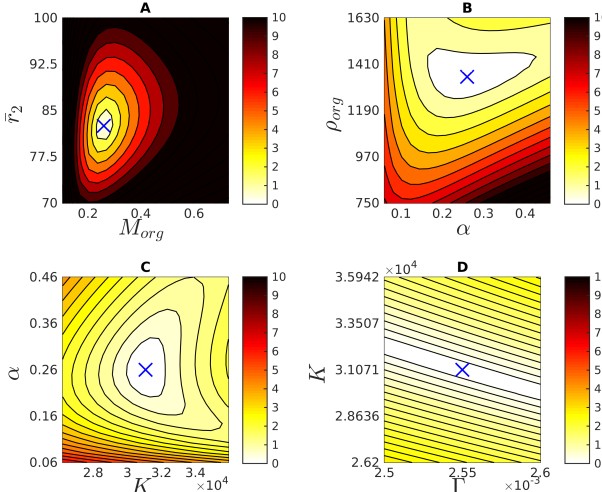

**Figure 9.** Response surfaces for $a_w^{np}\sigma^{nf}$ in the marine average environment. Blue crosses indicate the true parameter values $\theta^{true}$ used to calculate the calibration data. The colour scale represents the value of the OF calculated for the modelled spectra against the calibration data for parameter values across the uncertainty ranges (table 2). Contours identify constant OF values on the surface (note different scales and orders of magnitude).

Wex et al. (2008), in determining $S_c$. This effect manifests itself in increased CCN activity, Fig. 3 (red) for all environments when compared with simple Köhler theory using surface tension for water (blue circles). For this reason it is an issue that these parameters are not well researched, documented or constrained for many atmospherically relevant compounds - values can only be found for a handful of species (Topping, 2010). In addition, the parameter plane for the partitioning parameter Fig.

9D also shows a strong interaction between the two parameters that can result in non-unique optimised parameter values for a zero OF. This suggests that the OF requires a higher information content to correctly optimise $K$ and $\Gamma$ for this partitioning scheme.

### 4.2.4  The complete partitioning scheme: $a_w^p\sigma^{nf}$

Here the full partitioning framework is considered. The surface tension calculated using the partitioning parameters $\Gamma$ and $K$

as in section 4.2.3, we therefore have the same number of parameters for consideration and surfaces 10 of which possess a well defined minimum.

Figure 10 contains response surfaces for this comprehensive partitioning scheme. There are clearly strong similarities with Fig. 8C and D and once again the partitioning parameters seem to be relatively insensitive in contrast to $a_w^{np}\sigma^{nf}$. This is explained by the dependence of surface tension on the organic activities as well the partitioning parameters. For the $a_w^{np}\sigma^{nf}$

scheme there was substantial depression of surface tension through $\Gamma$ and $K$ and therefore significant sensitivity, here however the dependence on the organic activity pushes the value of surface tension back close to that of water at the point of activation.



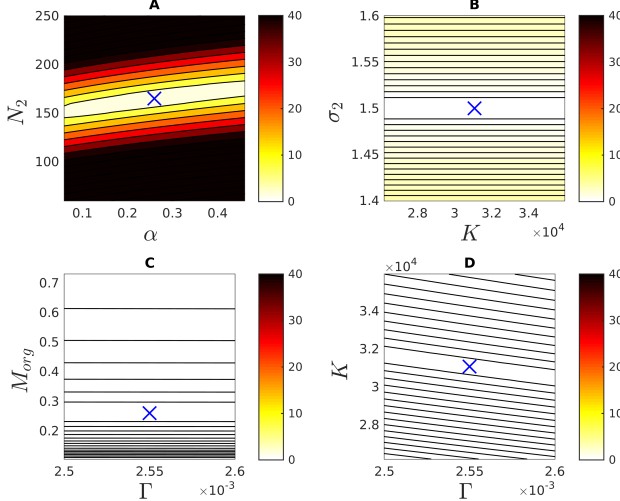

**Figure 10.** Response surfaces for $a_w^p \sigma^{nf}$ in the marine average environment. Blue crosses indicate the true parameter values $\theta^{true}$ used to calculate the calibration data. The colour scale represents the value of the OF calculated for the modelled spectra against the calibration data for parameter values across the uncertainty ranges (table 2). Contours identify constant OF values on the surface (note different scales and orders of magnitude).

This is also clear from the calibration data plotted in 3 (pink and blue). Here the same result has been extended over a range of partitioning parameter values using response surfaces.

The ability of simple Köhler theory, when the surface tension of water is used, to approximately replicate the CCN concentrations generated from the full partitioning treatment is in agreement with existing literature (Prisle et al., 2012, 2010;
Sorjamaa et al., 2004)

For the full partitioning scheme considered here, the qualitative relative sensitivity of each parameter, and both their linear and non-linear interactions, are summarised in table 3. Parameters that are indicated to have high or very high sensitivities are good candidates for a future study using automated search algorithms to provide a quantitative GSA and parameter calibration.

### 4.2.5 Environmental considerations

Sensitivities of the organic chemical parameters were not found to vary a significant amount between environments and therefore we have not included response surfaces for all environments in sections 4.2.1- 4.2.4. However, sensitivity of CCN spectra to size distribution parameters and $\alpha$ did show some environmental dependence, as one would expect. In Fig. 11A-C the response surfaces for $N_2$ and $\alpha$ perturbations are shown for all three environments. In panels A (marine average) and B (rural continental) very similar parameter interactions are evident - a zero value OF is returned across the entire range of uncertainty
in $\alpha$, suggesting $\alpha$ is insensitive and thus cannot be calibrated to a unique value based on the information content of the CCN spectrum alone for these environments. In panel C (polluted continental), a higher degree of sensitivity to $\alpha$ is clear from a



steeper gradient parallel to its axis. The increased sensitivity allows the uncertainty to be constrained subject to interactions with $N_2$ in more polluted environments. However, this constrained uncertainty in $\alpha$ has come at the cost of a less well defined $N_2$. It should be noted however, that despite this challenging outlook, response surfaces provide only a glimpse of the full parameter space and a well defined minimum may exist where a third, or several parameters, push CCN spectra into a

5    different sensitivity regime. To perform a rigorous analysis, automatic search algorithms must be employed using a selection of parameters believed to be identifiable from a thorough response surface analysis as presented here.

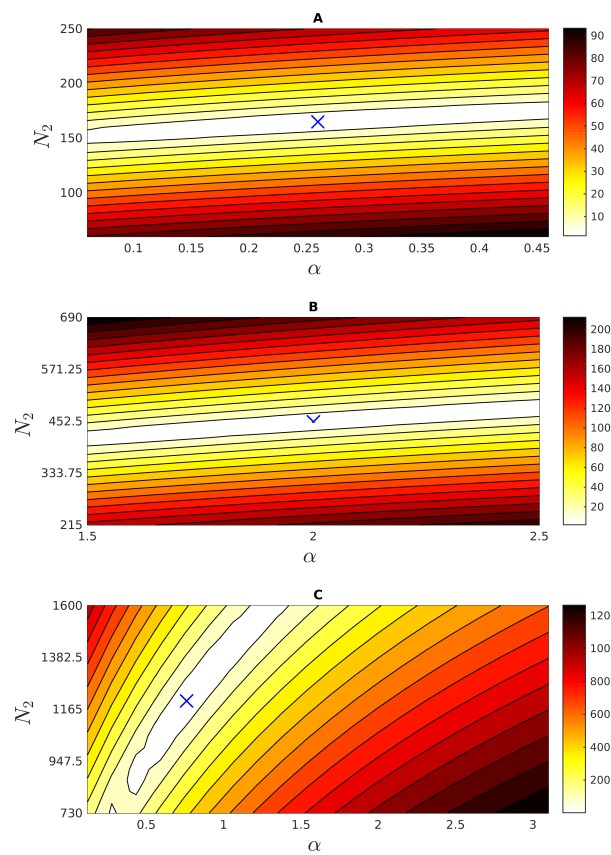

**Figure 11.** $N_2$ vs. $\alpha$ response surfaces of the OF for the $a_w^p \sigma^{nf}$ partitioning scheme for marine average (A), rural continental (B) and polluted continental (C) environments.

## 4.3   Implications for using CCN observations as calibration data

For a single parameter to be considered as identifiable only one response surface that shows a well-defined minimum is required. However, if the parameter does not exhibit well defined minima in several parameter pairs, and in particular if these

10   surfaces are relatively flat, then automatic search algorithms will likely struggle to converge on unique parameters values. This





indicates that such parameters are highly insensitive and thus accurate calibration is unnecessary for the model considered herein; in addition given the synthetic observations used in this study calibration will not be successful. In GCMs parameters such as surface tension and hygroscopicity parameter $\kappa$ (Petters and Kreidenweis, 2007) are often implemented as fixed values. Ervens et al. (2010) showed a sensitivity of 10% - 20% in cloud droplet number concentrations for perturbations in $\kappa$ over

several locations for updrafts typical of stratiform cloud types. Studies aiming to calibrate these GCM parameters that are not measurable in-situ at the scale of interest must therefore be cautious when selecting calibration data so as to include as much information content as possible. In this study it has been shown that using CCN spectra alone as calibration data will likely not be sufficient for calibration of all relevant parameters. Therefore, it is recommended that further studies are required to identify the appropriate in-situ measurements required and thus inform the experimentalist community accordingly. One way to help

abate this issue is to supply the objective function with a greater information content. In this study we have added a considerable amount of information content to CCN spectra already by using the interpolation methods outlined in section 2.4. Without these interpolation methods CCN spectra would certainly have a 'step-like' shape which would result in multiple local minima in response surfaces that would in general also be 'spiky'. When adding information content in a synthetic study such as the one performed here, it is important to be mindful that such information content could be retrieved from field observations as the

end goal is to compare with an observational data set rather than synthetic measurements. In this study the information content is solely composed of the deviation of model predictions for a single set of calibration data for each partitioning scheme and environment. If, in a future study, parameter search algorithms do not converge in an efficient manner additional information content can be supplied from in-flight temporal CCN measurements taken by counters at fixed supersaturations, or from the interstitial fraction of the the aerosol size distribution.

## 5 Conclusions

A methodology that is able to scrutinize the sensitivity of Köhler theory to perturbations in physicochemical parameters across a range of atmospherically relevant supersaturations has been constructed. The response surface tool allows us to perform such analysis of two such parameters simultaneously, thus probing a greater volume of parameter space than in standard one-at-a-time parameter sensitivity analyses. Across all partitioning schemes and environments a total of 543 response surfaces were

calculated.

In agreement with Djikaev and Ruckenstein (2014), the response surface analysis here confirms that the microscopic chemical properties of the organic compound have only a small effect on aerosol activation. In further agreement, Wex et al. (2007) parametrised their Köhler model with a bulk chemical parameter that contained the density, molecular weight and effective dissociation of the organic substance, and found it to take a constant value over a range of $M_{org}$ and $\rho_{org}$ values resulting in a

constant activation point.

For all partitioning cases, sensitivity to surface tension and the relative amount of organic mass present in the aerosol is on the order of that of size distribution parameters. This is in contradiction to Dusek et al. (2006) and certainly warrants further investigation. The ability of this novel methodology to probe sensitivity over a range of atmospherically relevant supersaturation



is likely the source of this difference as considerations in Dusek et al. (2006) are at fixed supersaturations. As response surfaces have shown relative similarities in sensitivity of CCN spectra to surface tension, relative organic content and size distribution parameters, an inverse modelling study using MCMC algorithmic methods will form the focus of a future study to better quantify these contrasting conclusions.

It has been deduced that compensating partitioning effects between bulk-surface partitioning and surface tension can generally be attributed to the changes in surface tension as the surface tension is also dependent upon the concentration gradient. This result is in agreement with global simulation performed by Prisle et al. (2012). Thus, the full treatment of bulk-surface partitioning returns CCN concentrations almost identical to those calculated using simple Köhler theory with a surface tension value of water. In contrast, Nozière et al. (2014) used state of the art extraction techniques (Baduel et al., 2012) that allowed

them to observe surface tension values as low as $30 \text{mNm}^{-2}$ in atmospheric aerosols when allowing for equilibration times. The importance of this is two-fold: Firstly, CCN spectra calculated using the complete partitioning model were a consequence of surface tension values close to that of water. In light of the measurements recorded by Nozière et al. (2014) this might suggest the bulk-surface framework might not be correct model for all surfactants behaviour. This warrants further theoretical and/or experimental investigations. Secondly, a future aerosol CCN closure study based on the framework built here requires careful

considerations if using existing experimental techniques; residence times for CCN counters, for example, are not long enough to facilitate such surfactant equilibration, therefore a closure study will likely prove unsuccessful or, at best, successful for the wrong reasons. Similarly, it has been suggested that lack of control of the saturation ratio of all semi-volatiles in such instruments might influence retrieved single particle properties (Topping and McFiggans, 2012). With respect to future use of the four partitioning schemes employed here, while treatment of surface tension depression or water activity alone lacks a

physically justified basis, we should not necessarily continue to use a simple Köhler theory with the surface tension of water or, at additional computational demand, the full treatment blindly, as neither formulations have seen adequate verification from experimental data relating to surfactants. This is complicated by the possibility of additional composition dependent processes not only related to surfactant behaviour. Nonetheless, in order to increase current understanding of the role that bulk-surface partitioning plays in cloud nucleation, development of more sophisticated instrumentation, such as that of Baduel et al. (2012),

must be a priority. In particular, CCN counters that operate on a highly resolved range of atmospheric supersaturations and have residence times to sufficiently capture equilibrium of surfactants are required. In addition, the use of single particle levitation techniques such as those used by Lienhard et al. (2015) might provide additional insights into the role of surface tension should they be able to access complex mixed aerosol.

     Ervens et al. (2005) examined several chemical and compositional effects simultaneously and found compensating parame-

ters resulted in a decreased sensitivity of total cloud droplet numbers when compared to studies treating the effects individual. Therefore, to have a good understanding of these effects global sensitivity analyses (GSA) are required. Response surfaces have here shown that several parameters may be identifiable in the complete parameter space and also that there are many interacting parameter pairs. Interacting parameters indicate the model under consideration can be simplified by reducing interacting parameters in single parameter as performed for $\kappa$ by Kreidenweis et al. (2005). Therefore the inverse modelling framework

developed here will be revisited in a future study. By implementing a Monte Carlo Markov Chain (MCMC) algorithm in a




similar manner as performed by Partridge et al. (2012), a statistically conditioned parameter optimisation and GSA can be conducted. The applicability of the algorithm will be first benchmarked against synthetic measurement data, i.e. calibration data used in this study, before being applied to real world measurements taken from the European Integrated project on Aerosol Cloud Climate and Air Quality interactions (EUCAARI) (Paramonov et al., 2015). While the response surface analysis here

suggests that a proper treatment of bulk-surface partitioning produces CCN concentrations similar to those of the classic Köhler theory, thus questioning its use in already computationally demanding global modelling, it only provides insight into 2D planes of the full parameter space. Using an MCMC simulation this preliminary conclusion can readdressed using a more rigorous approach that also provides a greater understanding of the entire parametric landscape.

At this stage, results documented collectively in table 3 show that there are many parameter interactions present in CCN

modelling. In addition, it is also clear that log-normal distribution parameters, compositional fractions, surface tension and solution ideality are all parameters that exhibit high sensitivity and as a community we must seek to reduce uncertainties in these parameters for effective global climate modelling.

*Acknowledgements.* This work was supported by the UK Natural Environment Research Council grants NE/I020148/1 (Aerosol-Cloud Interactions - A Directed Programme to Reduce Uncertainty in Forcing) and NE/J024252/1 (Global Aerosol Synthesis And Science Project).

P. Stier would like to acknowledge funding from the European Research Council under the European Union's Seventh Framework Programme (FP7/2007-2013) ERC project ACCLAIM (grant agreement no. FP7-280025).





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



| Component | $\rho$ (kgm$^{-3}$) | $M$ (gmol$^{-1}$) | Mass fractions | | |
|---|---|---|---|---|---|
| | | | Marine average | Polluted continental | Rural continental |
| Model organic | 1350 | 260 | 0.18 | 0.40 | 0.60 |
| BC | 2000 | 12 | 0.30 | 0.075 | 0.10 |
| $(NH_4)_2SO_4$ | 1770 | 132 | 0.00 | 0.2625 | 0.15 |
| NaCl | 2160 | 58.44 | 0.52 | 0.00 | 0.00 |
| $NH_4NO_3$ | 1720 | 80.55 | 0.00 | 0.2625 | 0.15 |

Table 1: Density, molecular weight and mass fraction of each aerosol component in all environments. The mass fractions included here are used to derive true parameter values for $f_{insol}$ and $\alpha$ in table 2.

| Environment | Marine Average | | | Polluted Continental | | | Rural Continental | | |
|---|---|---|---|---|---|---|---|---|---|
| Parameter | Min | True | Max | Min | True | Max | Min | True | Max |
| $\rho_{org}$ (kgm$^{-3}$) | 750 | 1350 | 1630 | 750 | 1350 | 1630 | 750 | 1350 | 1630 |
| $M_{org}$ (gmol$^{-1}$) | 105 | 260 | 730 | 105 | 260 | 730 | 105 | 260 | 730 |
| $\Phi$ | 0.75 | 1.0 | 1.0 | 0.75 | 1.0 | 1.0 | 0.75 | 1.0 | 1.0 |
| $\sigma$ (mNm$^{-2}$) | 30.0 | 55.0 | 72.8 | 30.0 | 55.0 | 72.8 | 30.0 | 55.0 | 72.8 |
| $\alpha$ | 0.06 | 0.26 | 0.46 | 0.12 | 0.76 | 3.10 | 1.50 | 2.00 | 2.50 |
| $f_{insol}$ | 0.10 | 0.30 | 0.50 | 0.03 | 0.075 | 0.12 | 0.05 | 0.10 | 0.15 |
| $K$ | 26200 | 31071 | 35942 | 26200 | 31071 | 35942 | 26200 | 31071 | 35942 |
| $\Gamma$ (mmolm$^{-2}$) | 0.0025 | 0.00255 | 0.0026 | 0.0025 | 0.00255 | 0.0026 | 0.0025 | 0.00255 | 0.0026 |
| $N_1$(cm$^{-3}$) | - | 265.00 | - | - | 4900.00 | - | - | 1010.00 | - |
| $\sigma_1$ | - | 1.45 | - | - | 1.55 | - | - | 1.71 | - |
| $\bar{r}_1$(nm) | - | 21.00 | - | - | 33.00 | - | - | 23.70 | - |
| $N_2$(cm$^{-3}$) | 60.00 | 165.00 | 250.00 | 730.00 | 1200.00 | 1600.00 | 215.00 | 451.00 | 690.00 |
| $\sigma_2$ | 1.40 | 1.50 | 1.60 | 1.50 | 1.55 | 1.62 | 1.40 | 1.58 | 1.75 |
| $\bar{r}_2$(nm) | 70.00 | 82.50 | 100.00 | 75.00 | 93.50 | 105.00 | 75.00 | 89.80 | 105.00 |

Table 2: True parameter values used for calibration data for all environments and their corresponding parameter ranges used for perturbations in the response surface analysis.



| Parameter | Relative sensitivity | Linear interactions | Non-linear interactions |
|-----------|---------------------|---------------------|-------------------------|
| $N_2$ | Very high | $\alpha$, $\Phi$ | $f_{insol}$ |
| $\sigma_2$ | Medium | - | - |
| $\bar{r}_2$ | High | - | - |
| $\alpha$ | High | $N_2$, $\Phi$ | $\rho_{org}$, $f_{insol}$ |
| $f_{insol}$ | High | - | $N_2$, $\rho_{org}$, $\alpha$, $\Phi$ |
| $K$ | Low | $\Gamma$ | - |
| $\Gamma$ | Low | $K$ | - |
| $\Phi$ | High | $N_2$, $\alpha$ | $f_{insol}$ |
| $\rho_{org}$ | Medium | - | $\alpha$, $f_{insol}$ |
| $M_{org}$ | Low | - | - |
| $\sigma$ | Very high | $\sigma_2$ | $M_{org}$, $\rho_{org}$, $\alpha$, $f_{insol}$, $\Phi$ |

Table 3: Summary of qualitative sensitivities and parameter interactions observed in response surfaces for all parameters used in the the complete partitioning scheme $a_w^p \sigma^{nf}$ for the marine environment. The surface tension $\sigma$ for classical Köhler theory $a_w^{np} \sigma^f$ is also included at the bottom of the table.