# Peer review of "Inverse modelling of Köhler theory - Part 1: A response surface analysis of CCN spectra with respect to surface-active organic species"

_Atmospheric Chemistry and Physics, 2015_

## Referee Comment (RC1) · Anonymous Referee #1 · 11 Mar 2016

The authors present a new approach to assess the sensitivities of various parameters to CCN numbers. As opposed to numerous previous studies that used a 'one-at-a-time' approach, i.e. varying only one parameter at once, their use of response surfaces can reveal sensitivities over much wider parameter spaces. The focus on surface partitioning and surface tension and it is concluded that careful parameterization of these parameters is needed in order to successfully predict CCN under some conditions, in agreement with previous studies. The current study represents a model framework and the input data are artificially created so that perfect agreement can be achieved. So, therefore it is quite simplistic and does not lead to entirely new results. The sensitivity study for the selected parameters is more comprehensive and the approach might

be promising to be applied to other problems in the future. However, not all parameters that have been identified as being important for determining the CCN numbers in previous studies have been discussed. Therefore, I think the paper's content and conclusions are somewhat weak and preliminary since only the suitability of the model framework, but not many new results are discussed.

Major comments

1) Mixing state In several previous studies, it has been emphasized that the mixing state of aerosol particles might be one of the most important parameters that determines CCN number in fresh air masses. While it has been addressed briefly in the manuscript, it should be discussed more thoroughly. Could a measure of mixing state be included in the model framework?

2) Previous study on CCN sensitivities In a previous study, Lee et al. (2013) have performed a sensitivity study on a global scale of many parameters using a Monte-Carlo-based approach. This study should be discussed in the light of the results in the current study.

3) Uncertainty in cloud formation The authors state correctly that for data sets other than their calibration data set as used here, a perfect agreement cannot be expected. Some discussion on how accurately CCN numbers should be predicted should be added. For example, all measurements are associated with some measurement error. In addition, other factors influence CCN number (e.g. Lee et al. (2013). Given all uncertainties in the current abilities to predict cloud formation (meteorology, updrafts, emissions, etc), what is the recommendation for a tolerable uncertainty in CCN predictions?

4) Solubility In previous studies, the solubility of organics has been identified as an influential parameter (Riipinen et al., 2015). However, this is not even mentioned in the current paper since all organics are assumed to be completely dissolved. How would consideration of a range of solubilities change the conclusions?

5) Surface-active fraction The discussion of previous studies on the effect of surface tension to CCN activation is somewhat confusing. For example, studies by Noziere et al are discussed that reveal surface tension of $\sim$ 30 nM on aerosols. This number, however, is not relevant for CCN activation unless sufficient material is available to cover the complete particle/'droplet near activation. I suggest discussing the cited references more carefully. What fraction of surface-active material is needed to cause the effects as seen in the current study? Are these fractions realistic?

6) All figures It is not clear to me why the authors do not show all response surfaces (as a supplement). If they choose not to do it, more emphasize should be given why the figures they are showing are shown and not others.

7) Figures Several figures appear blurry and hard to read due to very small font, e.g., Fig. 1 and 3: the indices in the legend are hard to distinguish Fig. 4: Avoid putting the legend box across the lines in the figure

Minor comments

Abstract: The fact that no 'real data' but an artificially created calibration data set has been used, should be mentioned in the abstract.

p. 3, l. 33 (and numerous other places): OF has been defined before

p. 4, l. 18: 'Köhler models' sounds rather colloquial

p. 4, l. 27: Since all the listed studies are model studies, 'observed' should be replaced by 'implied' (or something similar) as the results do not directly refer to observations

p. 5, l. 4: Add a reference for the fact that organic composition has changed since the preindustrial times.

p. 6, l. 11: Sounds odd. 'Is inverse modelling... a well posed problem?' as not the modelling is the problem but the sensitivity to CCN number

p. 11, l. 11: This sentence seems redundant (cf. l. 7)

Technical comments

p. 4, l. 20: supersaturations –> supersaturation

p. 8, l. 8: 'a' can be omitted

p. 9, l. 22: SRFA is. . . –> SRFA has. . . (?)

p. 14, l. 8: is –> are

p. 15, l. 17: can assessed –> can be assessed

p. 24, l. 1: in 3 –> in Fig. 3

References

Lee, L. A., Pringle, K. J., Reddington, C. L., Mann, G. W., Stier, P., Spracklen, D. V., Pierce, J. R., and Carslaw, K. S.: The magnitude and causes of uncertainty in global model simulations of cloud condensation nuclei, Atmos. Chem. Phys., 13, 17,8879-8914, 10.5194/acp-13-8879-2013, 2013.

Riipinen, I., Rastak, N., and Pandis, S. N.: Connecting the solubility and CCN activation of complex organic aerosols: a theoretical study using solubility distributions, Atmos. Chem. Phys., 15, 11,6305-6322, 10.5194/acp-15-6305-2015, 2015.

---

## Referee Comment (RC2) · Anonymous Referee #2 · 22 Mar 2016

This study presents a methodology to investigate the sensitivity of CCN spectra to different parameters using different approaches. The results shown are not entirely new and future work is necessary in order to relate them to "real world" characterized by simultaneous measurements of aerosol chemical composition and size distribution, supersaturation, CCN spectra, etc.

The theoretical investigations shown here were performed at a fixed temperature, using literature data for three aerosol types: marine, polluted continental and rural. The organic aerosol and surface tension were assumed to be the same for all aerosol types. The possible effects of this data are not discussed in the manuscript.

The manuscript looks like a report, most of the section Section 4.2 may be moved in

a Supplementary Material. The results shown in it have to be presented in a more synthetic and comparative manner. It is not evident how much the sensitivity to parameters depends on the approach used in a quantitative manner. This is also not clear in Table 3 which is not actually discussed. Also, it has to be moved from Conclusions to Results.

Such a study may be very interesting if it succeeds in showing the limits/differences due to the different approaches used in modelling aerosol activation and quantify the acceptable/relevant uncertainties of measured parameters of fundamental importance in these approaches. The latter information may be important for planning future field campaigns and for development of instruments.

Minor comments:

The term "posedness" is not commonly used, replace or explain it better.

---

## Author Comment (AC2) · 12 Jul 2016

This study presents a methodology to investigate the sensitivity of CCN spectra to different parameters using different approaches. The results shown are not entirely new and future work is necessary in order to relate them to "real world" characterized by simultaneous measurements of aerosol chemical composition and size distribution, supersaturation, CCN spectra, etc.

**The authors would like to take this opportunity to thank the reviewer for their constructive comments; we believe that they have added to the quality of the manuscript as a whole. The purpose of this study was to present the development of a new framework for probing sensitivity of aerosol activation to pro-**

cesses which have been studied in isolation over a number of years. By embracing an inverse modelling approach to aerosol-CCN closure, we not only build a framework for sensitivity analysis, but also a method of diagnosing both structural and parametric uncertainties in model CCN predictions by simultaneously matching input parameters and model output. In section 5.3 of the revised manuscript, additional material is presented to highlight the importance of calibration data resolution and natural variability in CCN measurements as a natural first step towards future work with observational data. Our response is presented in bold text following the reviewer's comments. Any referencing of sections, pages or line numbers given in the response pertain to those of the revised manuscript.

**Major comments:**

1. The theoretical investigations shown here were performed at a fixed temperature, using literature data for three aerosol types: marine, polluted continental and rural. The organic aerosol and surface tension were assumed to be the same for all aerosol types. The possible effects of this data are not discussed in the manuscript.

**Preliminary testing of temperature was performed and negligible sensitivity was found. We therefore chose to exclude it from our analysis. The reviewer raises a good point with regard to the organic aerosol data. Unfortunately, as the organic aerosol fraction exhibits such high levels of complexity and spatial and temporal variability, in order to be more precise than we have been is difficult. By providing generous ranges for our model organic we hope to encompass all possibilities that may be realised in the real atmosphere. Indeed, once coupled with the MCMC algorithm (part 2) the framework developed in the present study provides a useful tool to constrain such parametric uncertainties by exploring all possibilities in the complete parameter space.**

2. The manuscript looks like a report, most of the section 4.2 may be moved in a Supplementary Material. The results shown in it have to be presented in a more synthetic and comparative manner. It is not evident how much the sensitivity to parameters depends on the approach used in a quantitative manner. This is also not clear in Table 3 which is not actually discussed. Also, it has to be moved from Conclusions to Results.

**We agree that the manuscript could be more concise, therefore the text has been substantially reduced and formulated in a more synthetic and concise manner. A supplementary document that contains all response surface analysis has now been attached with the manuscript. However, as inverse modelling of CCN spectra has not been performed before, we believe that the response surface analysis contained in section 5.2 (previously 4.2) is essential to the understanding of how to correctly couple the framework to automatic search algorithms, such as MCMC, which will form the focus of a part 2 study. Therefore we choose to keep the section in the main text.**

3. Such a study may be very interesting if it succeeds in showing the limits/differences due to the different approaches used in modelling aerosol activation and quantify the acceptable/relevant uncertainties of measured parameters of fundamental importance in these approaches. The latter information may be important for planning future field campaigns and for development of instruments.

**The reviewer raises a very good point here and we have chosen to add additional material to section 5.3 to address the issue at hand. Therein we have discussed at length the importance of information content for various definitions of the calibration that may arise due to different instrumentation. We also account for the natural variability in these considerations. It was concluded that, when accounting for natural variability in the analysis, it would be challenging to correctly minimise the OF based on the information content of a calibration data set measured by a typical CCN counter. However, correct minimisation of the OF was still achieved when corrupting the high resolution calibration data set with randomly generated natural variability. As such, this result should serve**

[Figure]

**as a recommendation for the development of instrumentation for high resolution measurements of CCN spectra in-situ.**

**Minor comments:**

The term "posedness" is not commonly used, replace or explain it better

**The term posedness has been replaced throughout the manuscript with 'well-posed', 'ill-posed' or similar.**

---

## Author Response (AR1)

Dear Veli-Matti Kerminen,

We would like to thank both yourself and the referees for the time you have spent considering our study and also for your patience with regard to the generous extension we received. Owing to the constructive feedback provided by the referees, we believe the revisions and additions have added a lot of value to the manuscript. Please find the author responses, list of manuscript changes and the annotated, revised manuscript contained below in this document.

**Response to referees:**

Below are the authors' point-by-point responses to each of the referees' comments. General comments are presented before major comments which precede technical corrections/minor comments. Our responses follow (bold italic text) the referee comments (standard text).

Anonymous referee 1:

*General comments:*

The authors present a new approach to assess the sensitivities of various parameters to CCN numbers. As opposed to numerous previous studies that used a 'one-at-a-time' approach, i.e. varying only one parameter at once, their use of response surfaces can reveal sensitivities over much wider parameter spaces. The focus on surface partitioning and surface tension and it is concluded that careful parameterization of these parameters is needed in order to successfully predict CCN under some conditions, in agreement with previous studies. The current study represents a model framework and the input data are artificially created so that perfect agreement can be achieved. So, therefore it is quite simplistic and does not lead to entirely new results. The sensitivity study for the selected parameters is more comprehensive and the approach might be promising to be applied to other problems in the future. However, not all parameters that have been identified as being important for determining the CCN numbers in previous studies have been discussed. Therefore, I think the paper's content and conclusions are somewhat weak and preliminary since only the suitability of the model framework, but not many new results are discussed.

***The authors would like to take this opportunity to thank the reviewer for their constructive comments that we believe have added to the quality of the manuscript as a whole. The purpose of this study was to present a development of a new framework for probing sensitivity of aerosol activation to processes which have been studied in isolation over a number of years. The referee is absolutely right that there are a number of additional factors affecting CCN activation potential, but we focus on surfactant representations since not only is it a useful proof of concept demonstration of this framework, but there is still no clear message as to the wider relevance of surfactant behaviour. Using bulk to surface partitioning models, such as those presented here, previous studies have shown the impact of extreme behaviour, which one might consider as partitioning on or off, on the global scale, but suggested further studies would be better constrained by a wider parameter space such as provided here. In addition, as the referee points out, there are still studies promoting the potential for very low effective surface tension values, even if the concentration of surfactant material at activation might never be realised in the atmosphere. We feel demonstrating the use of a new multi-parameter sensitivity approach in helping to resolve the wider relevance of such issues is important, but requires incremental demonstrations of its use. We fully agree future studies need to tackle the issue of inter-instrument variability and process descriptions, but would also warrant much more data on systems for which we know the pure***

*component and mixture properties more accurately than ambient systems. We hope our response to the detailed points below make this clear, as do recommendations for future work.*

*Our responses are presented in bold text following the reviewer's comments. Any referencing of sections, pages or line numbers given in the response pertain to those of the revised manuscript.*

*Major comments:*

1. Mixing state In several previous studies, it has been emphasized that the mixing state of aerosol particles might be one of the most important parameters that determines CCN number in fresh air masses. While it has been addressed briefly in the manuscript, it should be discussed more thoroughly. Could a measure of mixing state be included in the model framework?

**This a good point, CCN activity may indeed be influenced by the mixing state close to aerosol sources, and as such additional text and references have been added to section 3.1. The framework developed here could in principle be used for a treatment of externally mixed aerosols, however, the choice of mechanistic aerosol-cloud model would be important. In theory, frameworks such as PartMC-MOSAIC (Tian et al 2014) could be coupled with our MCMC approach. Furthermore, the methodology developed here could be coupled with a with multi-modal cloud parcel model in the future. One of the virtues of the framework built in the present study is that many different aerosol processes and characteristics can be included in the future for more specific case studies.**

2. Previous study on CCN sensitivities In a previous study, Lee et al. (2013) have performed a sensitivity study on a global scale of many parameters using a MonteCarlo-based approach. This study should be discussed in the light of the results in the current study.

**The Lee et al. (2013) is indeed an interesting study. However, due to numerous differences between our study and that performed by Lee et al. we feel that the study is not directly of relevance.**

**Firstly, the studies have different goals. The focus of this study is to construct a framework, based on inverse modelling methods, for which model input parameters non-measurable at the scale of interest (e.g surface tension) can be calibrated against measurements of CCN spectra, and ultimately that parametric uncertainty can be evaluated and constrained using MCMC analysis (part2). In contrast, the study performed by Lee et al. (2013) is concerned only with sensitivity evaluation. In this part 1 study, by testing the validity of the inverse approach through response surfaces, we also get a visualisation of the model's parametric sensitivity in terms of Objective Function (OF) response surfaces.**

**It is well understood that application of inverse modelling methodologies, both for model calibration and parametric uncertainty analysis, can result in (near-)identical model predictions for non-unique parameter values if the system studied contains parameters that are non-identifiable for the definition of calibration data used. In such cases it can be challenging for the applied algorithm to converge on optimal parameter values in an attempt to constrain uncertainties. Typically, such difficulties are approached by increasing the information content in the calibration data (synthetic or real-world measurements) (Vrugt et al 2001). Therefore, we choose CCN spectra as calibration data in the present study to maximise the available information content with respect to currently known observations - as compared with the single N50 value used in Lee et al. (2013). Appropriate definition of calibration data and the importance of information content is now discussed at some length in section 5.3 of the revised manuscript as an appropriate definition of calibration data is essential for successful application of automatic search algorithms for parameter calibration and parametric uncertainty analysis. In section 5.3 we have explored the**

*implications of the information content contained within the calibration data for the identifiability of parameters investigated depending on the resolution of the calibration data, both with and without corruption of the synthetic measurements by a randomly generated natural variability. We conclude in the case of uncorrupted calibration data that a typical 5-band CCNC spectrum would contain sufficient information content for the presented methodology, but that it is unlikely that the use of a single value would facilitate parameter optimisation. When also accounting for natural variability, only the high-resolution calibration data, as used in the present study, would suffice for correct minimisation of the OF.*

*The dissimilarity in the definition of the calibration data between the present study and Lee et al (2013) is a natural consequence of another difference between the two studies – the scale. Our study is focused on developing a process model framework suitable for further uncertainty analysis in closure studies using algorithmic approaches, and we have referenced to similar studies accordingly Partridge et al. (2011, 2012). Such a study provides us with the opportunity to scrutinise the necessity for complexity, at a level of Köhler theory, offline whilst maintaining computational efficiency. Consideration of the impact of bulk atmospheric parameters (e.g. emission rates), as seen in Lee et al. (2013), does not fall under the remit of the present study and instead we choose to highlight the importance of Köhler model complexity in GCMs for the most accurate predictions of CCN.*

*To summarise, any sensitivity methodology relies heavily on the choice of calibration data. The focus of the present study is the construction of a framework for an inverse modelling approach to parametric uncertainty analysis and model calibration for entire CCN spectra at a process level rather than a single value on a global scale. Therefore, it is of our opinion that in some sense the present study is both more complex and robust in relation to potential model-observation evaluation and depth of process treatment. By performing this kind of analysis over a range of supersaturations the global climate modelling community can gain a lot from the end results.*

*Therefore, owing to the dissimilarities in methodology and purpose of the two studies, and that Monte-Carlo methods have not been employed here in part 1, direct comparison of results with respect to sensitivity is not possible and thus we have chosen to cite appropriately and not discuss Lee et al. (2013) at great length within the manuscript.*

3. Uncertainty in cloud formation The authors state correctly that for data sets other than their calibration data set as used here, a perfect agreement cannot be expected. Some discussion on how accurately CCN numbers should be predicted should be added. For example, all measurements are associated with some measurement error. In addition, other factors influence CCN number (e.g. Lee et al. (2013). Given all uncertainties in the current abilities to predict cloud formation (meteorology, updrafts, emissions, etc), what is the recommendation for a tolerable uncertainty in CCN predictions?

*The reviewer makes an excellent point regarding the treatment of errors and acceptable uncertainty. Following a similar request from reviewer 2, additional material has been added to the manuscript in section 5.3. Therein we have discussed at length the importance of information content for various definitions of the calibration that may arise due to different instrumentation. We also account for the natural variability in these considerations. It was concluded that, when accounting for natural variability in the analysis, the typical supersaturation resolution of CCN counters used in-situ would be challenging to correctly minimise the objective function as done when using the high-resolution spectra as seen in the present study. As natural variability in CCN measurements is typically on the order of tens of percent this effect will dominate over any instrumentation errors introduced, counting errors as deduced from Poisson statistics, for example. Thus we choose to focus on natural variability in this study to illustrate this point.*

4. Solubility In previous studies, the solubility of organics has been identified as an influential parameter (Riipinen et al., 2015). However, this is not even mentioned in the current paper since all organics are assumed to be completely dissolved. How would consideration of a range of solubilities change the conclusions?

*We agree that solubility may play a role, as will mixing state, gas to particle partitioning, condensed phase reactions and mass transfer limitations according to an amorphous condensed phase, depending on the RH history of the particle distribution in question. In this framework, the solubility would change the single particle hygroscopicity that would need to be constrained by an equivalent parameter space to the one presented here. This would need to be coupled to an appropriate gas phase mechanism since the solubility spectrum, such as that presented by Riipinen et al (2015), is driven by the specific abundance of compounds within different volatility ranges. We would argue then that any treatment of solubility should also be coupled to a model that can treat gas to particle partitioning since the most thermodynamically stable state of a framework that does not allow partitioning might never be met, affecting the derived parameter sensitivity. Such a model framework is beyond the scope of this study, but for sure warrants future investigation. Again, in theory this framework could be wrapped around any mechanistic or semi-empirical model, the results of this stepwise study demonstrating its use.*

5. Surface-active fraction The discussion of previous studies on the effect of surface tension to CCN activation is somewhat confusing. For example, studies by Noziere et al are discussed that reveal surface tension of ~30 nM on aerosols. This number, however, is not relevant for CCN activation unless sufficient material is available to cover the complete particle/'droplet near activation. I suggest discussing the cited references more carefully. What fraction of surface-active material is needed to cause the effects as seen in the current study? Are these fractions realistic?

*The reviewer has raised a very good point here. This issue is relevant for Köhler frameworks 1 (traditional Köhler theory) and 2 (redistribution of surfactant concentration and a concentration-independent fixed surface tension).*

*One of the key results of this paper is that the complete partitioning framework (4) produces CCN spectra and response surfaces similar to traditional Köhler theory as a consequence that surface tension is often very close to that of water at the point of activation and largely insensitive to the partitioning parameters used in the Szyszkowski equation. While this result is perhaps well known in the communities of the references in question, it is arguably less well known in the cloud physics community. Text relating to the references has been reformulated and revised, and the result has been discussed more carefully.*

6. Figures Several figures appear blurry and hard to read due to very small font, e.g., Fig. 1 and 3: the indices in the legend are hard to distinguish Fig. 4: Avoid putting the legend box across the lines in the figure

*We apologise for this – these figures have now been remedied.*

*Technical corrections/minor comments:*

*All minor comments have been addressed, as advised, in the revised manuscript. Thank you for taking the time to bring these to our attention.*

*The reviewer raises a very good point here and we have chosen to add additional material to section 5.3 to address the issue at hand. Therein we have discussed at length the importance of information content for various definitions of the calibration that may arise due to different instrumentation. We also account for the natural variability in these considerations. It was concluded that, when accounting for natural variability in the analysis, it would be challenging to correctly minimise the OF based on the information content of a calibration data set measured by a typical CCN counter. However, correct minimisation of the OF was still achieved when corrupting the high resolution calibration data set with randomly generated natural variability. As such, this result should serve as a recommendation for the development of instrumentation for high resolution measurements of CCN spectra in-situ.*

*Technical corrections/minor comments:*

The term "posedness" is not commonly used, replace or explain it better

*The term posedness has been replaced throughout the manuscript with 'well-posed', 'ill-posed' or similar.*

**Changes to manuscript:**

In order to increase the quality of the manuscript and satisfy the referees considerable revisions and additions have been made to the manuscript. In particular, a lot of text has been rephrased and some methodology sections retitled and reordered to improve the readability of the study. In most sections content has remained unchanged, however, there have been considerable additions to section 5.3 which we feel have added a new perspective to the paper. As such, results contained therein have also been carried over to the abstract and conclusions. Below are a list of changes made to the figures and text, and in the marked up manuscript that follows we have highlighted new text and text that has been reworded considerably in yellow. References that have been added during the revision period, some at the request of the referees and others independently, have been highlighted in green. We hope that the changes the manuscript has undergone are clear from this.

Figures:

1. All figures have been updated for aesthetic purposes and as a minor bug was found in the setup of the size distributions. Fortunately, the bug had no impact on the results or conclusions reached by the study.

2. The environment presented in figure 5 (previously 4) has been changed from rural continental to marine average, and mode size has also been included (orange), to link with new findings in section 5.3 (previously 4.3).

3. Two additional figures have been added to section 5.3 (previously 4.3), figures 12 and 13.

Text:

1. New results reached in section 5.3 have been outlined in the first paragraph of the abstract. Results detailed previously in the abstract have been made more concise.

2. Contents of the introduction have remained unchanged. However, to improve the readability of the introduction, the ordering has changed substantially.

3. The goals of the study (section 1.1) have been rephrased

4. A considerable amount of text has been added to section 5.3 (previously 4.3) and pre-existing text reformulated.

5. Conclusions pertaining to results in section 5.3 have now have been added to the conclusions.

[revised manuscript text omitted]